# B-Spline CNNs on Lie Groups

**Erik J. Bekkers** [*]

| | | |
|---|---|---|
| Amsterdam Machine Learning Lab | · | Centre for Analysis and Scientific Computing |
| Informatics Institute | · | Applied Mathematics and Computer Science |
| University of Amsterdam | · | Eindhoven University of Technology |

`e.j.bekkers@uva.nl`

## Abstract

Group convolutional neural networks (G-CNNs) can be used to improve classical CNNs by equipping them with the geometric structure of groups. Central in the success of G-CNNs is the lifting of feature maps to higher dimensional disentangled representations in which data characteristics are effectively learned, geometric data-augmentations are made obsolete, and predictable behavior under geometric transformations (equivariance) is guaranteed via group theory. Currently, however, the practical implementations of G-CNNs are limited to either discrete groups (that leave the grid intact) or continuous compact groups such as rotations (that enable the use of Fourier theory). In this paper we lift these limitations and propose a modular framework for the design and implementation of *G-CNNs for arbitrary Lie groups*. In our approach the differential structure of Lie groups is used to expand convolution kernels in a generic basis of B-splines that is defined on the Lie algebra. This leads to a flexible framework that enables *localized*, *atrous*, and *deformable convolutions* in G-CNNs by means of respectively *localized*, *sparse* and *non-uniform B-spline* expansions. The impact and potential of our approach is studied on two benchmark datasets: cancer detection in histopathology slides in which rotation equivariance plays a key role and facial landmark localization in which scale equivariance is important. In both cases, G-CNN architectures outperform their classical 2D counterparts and the added value of atrous and localized group convolutions is studied in detail.

## 1 Introduction

Group convolutional neural networks (G-CNNs) are a class of neural networks that are equipped with the geometry of groups. This enables them to profit from the structure and symmetries in signal data such as images (Cohen & Welling, 2016). A key feature of G-CNNs is that they are equivariant with respect to transformations described by the group, i.e., they guarantee predictable behavior under such transformations and are insensitive to both local and global transformations on the input data. Classical CNNs are a special case of G-CNNs that are equivariant to translations and, in contrast to unconstrained NNs, they make advantage of (and preserve) the basic structure of signal data throughout the network (LeCun et al., 1990). By considering larger groups (i.e. considering not just translation equivariance) additional geometric structure can be utilized in order to improve performance and data efficiency (see G-CNN literature in Sec. 2).

Part of the success of G-CNNs can be attributed to the lifting of feature maps to higher dimensional objects that are generated by matching kernels under a range of poses (transformations in the group). This leads to a disentanglement with respect to the pose and together with the group structure this enables a flexible way of learning high level representations in terms of low-level activated neurons observed in specific configurations, which we conceptually illustrate in Fig. 1. From a neuro-psychological viewpoint, this resembles a hierarchical composition from low- to high-level features akin to the recognition-by-components model by Biederman (1987), a viewpoint which is also adopted in work on capsule networks (Hinton et al., 2011; Sabour et al., 2017). In particular in (Lenssen et al., 2018) the group theoretical connection is made explicit with equivariant capsules that provide a sparse index/value representation of feature maps on groups (Gens & Domingos, 2014).

---

[*] Work done at Eindhoven University of Techology

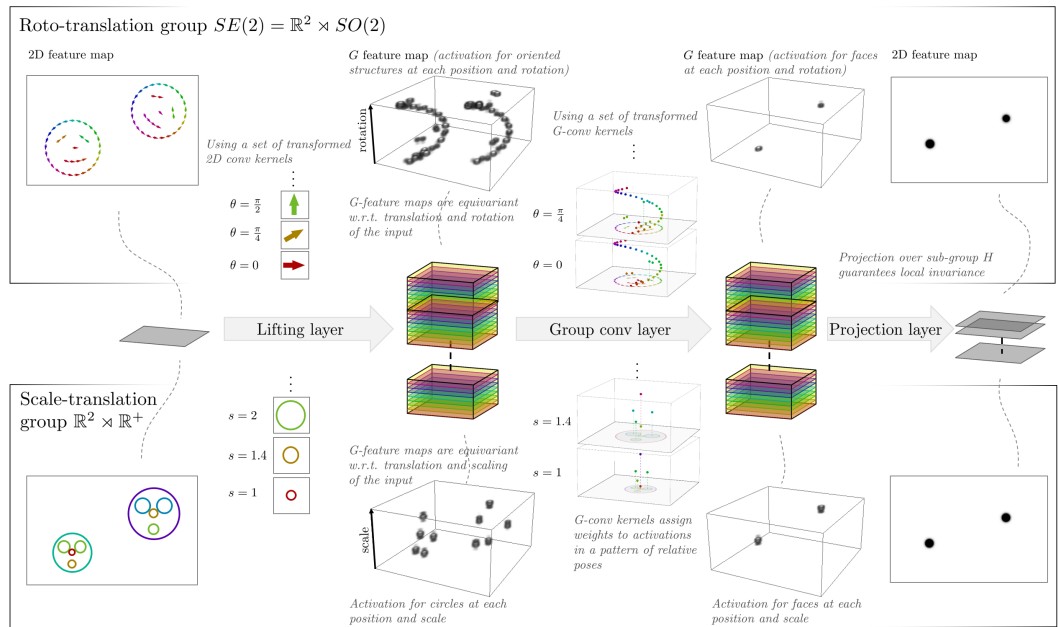

Figure 1: In G-CNNs feature maps are lifted to the high-dimensional domain of the group $G$ in which features are disentangled with respect to pose/transformation parameters. G-convolution kernels then learn to recognize high-level features in terms of patterns of relative transformations, described by the group structure. This is conceptually illustrated for the detection of faces, which in the $SE(2)$ case are considered as a pattern of lines in relative positions and orientations, or in the $\mathbb{R}^2 \rtimes \mathbb{R}^+$ case as blobs/circles in relative positions and scales.

Representing low-level features via features maps on groups, as is done in G-CNNs, is also motivated by the findings of Hubel & Wiesel (1959) and Bosking et al. (1997) on the organization of orientation sensitive simple cells in the primary visual cortex V1. These findings are mathematically modeled by sub-Riemannian geometry on Lie groups (Petitot, 2003; Citti & Sarti, 2006; Duits et al., 2014) and led to effective algorithms in image analysis (Franken & Duits, 2009; Bekkers et al., 2015b; Favali et al., 2016; Duits et al., 2018; Baspinar, 2018). In recent work Montobbio et al. (2019) show that such advanced V1 modeling geometries emerge in specific CNN architectures and in Ecker et al. (2019) the relation between group structure and the organization of V1 is explicitly employed to effectively recover actual V1 neuronal activities from stimuli by means of G-CNNs.

G-CNNs are well motivated from both a mathematical point of view (Cohen et al., 2018a; Kondor & Trivedi, 2018) and neuro-psychological/neuro-mathematical point of view and their improvement over classical CNNs is convincingly demonstrated by the growing body of G-CNN literature (see Sec. 2). However, their practical implementations are limited to either discrete groups (that leave the grid intact) or continuous, (locally) compact, unimodular groups such as roto-translations (that enable the use of Fourier theory). In this paper we lift these limitations and propose a framework for the design and implementation of *G-CNNs for arbitrary Lie groups*.

The proposed approach for G-CNNs relies on a definition of B-splines on Lie groups which we use to expand and sample group convolution kernels. B-splines are piece-wise polynomials with local support and are classically defined on flat Euclidean spaces $\mathbb{R}^d$. In this paper we generalize B-splines to Lie groups and formulate a definition using the differential structure of Lie groups in which B-splines are essentially defined on the (flat) vector space of the Lie algebra obtained by the logarithmic map, see Fig. 2. The result is a flexible framework for B-splines on arbitrary Lie groups and it enables the construction of G-CNNs with properties that cannot be achieved via traditional Fourier-type basis expansion methods. Such properties include *localized*, *atrous*, and *deformable convolutions* in G-CNNs by means of respectively *localized*, *sparse* and *non-uniform B-splines*.

Although concepts described in this paper apply to arbitrary Lie groups, we here concentrate on the analysis of input data that lives on $\mathbb{R}^d$ and consider G-CNNs for affine groups $G = \mathbb{R}^d \rtimes H$

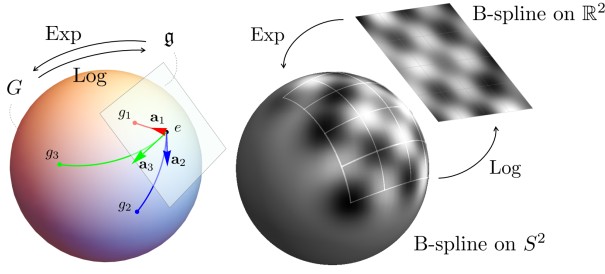

Figure 2: The Log-map allows us to map elements from curved manifolds such as the 2-sphere to a flat Euclidean tangent space. For Lie groups the Log-map is analytic, globally defined, and it provides us with a flexible tool to define group convolution kernels via B-splines. In our Lie group context the 2-sphere is treated as a quotient group $SO(3)/SO(2)$. Technical details are given in Sec. 3 and App. B.

that are the semi-direct product of the translation group with a Lie group $H$. As such, only a few core definitions about the Lie group $H$ (group product, inverse, Log, and action on $\mathbb{R}^d$) need to be implemented in order to build full G-CNNs that are locally equivariant to the transformations in $H$.

The impact and potential of our approach is studied on two datasets in which respectively rotation and scale equivariance plays a key role: cancer detection in histopathology slides (PCam dataset) and facial landmark localization (CelebA dataset). In both cases G-CNNs out-perform their classical 2D counterparts and the added value of atrous and localized G-convolutions is studied in detail.

## 2 RELATED WORK

**G-CNNs** The introduction of G-CNNs to the machine learning community by Cohen & Welling (2016) led to a growing body of G-CNN literature that consistently demonstrates an improvement of G-CNNs over classical CNNs. It can be roughly divided into work on discrete G-CNNs (Cohen & Welling, 2016; Dieleman et al., 2016; Winkels & Cohen, 2018; Worrall & Brostow, 2018; Hoogeboom et al., 2018), regular continuous G-CNNs (Oyallon & Mallat, 2015; Bekkers et al., 2015a; 2018b; Weiler et al., 2018a; Zhou et al., 2017; Marcos et al., 2017) and steerable continuous G-CNNs (Cohen et al., 2018b; Worrall et al., 2017; Kondor & Trivedi, 2018; Thomas et al., 2018; Weiler et al., 2018a; Esteves et al., 2018a; Andrearczyk et al., 2019). Since 3D rotations can only be sampled in very restrictive ways (without destroying the group structure) the construction of 3D roto-translation G-CNNs is limited. In order to avoid having to sample all together, steerable (G-)CNNs can be used. These are specialized G-CNNs in which the kernels are expanded in circlar/spherical harmonics and computations take place using the basis coefficients only (Chirikjian & Kyatkin, 2000; Franken, 2008; Almsick, 2007; Skibbe & Reisert, 2017). The latter approach is however only possible for unimodular groups such as roto-translations.

**Scale equivariance** In this paper we experiment with scale-translation G-CNNs, which is the first direct application of G-CNNs to achieve equivariance beyond roto-translations. Scale equivariance is however addressed in several settings (Henriques & Vedaldi, 2017; Esteves et al., 2018b; Marcos et al., 2018; Tai et al., 2019; Worrall & Welling, 2019; Jaderberg et al., 2015; Li et al., 2019), of which (Worrall & Welling, 2019) is most related. There, scale-space theory and semi-group theory is used to construct scale equivariant layers that elegantly take care of moving band-limits due to rescaling. Although our work differs in several ways (e.g. non-learned lifting layer, discrete group convolutions via atrous kernels, semi-group theory), the first two layers of deep scale-space networks relate to our lifting layer by treating our B-splines as a superposition of dirac deltas transformed under the semi-group action of (Worrall & Welling, 2019), as we show in App. C.1. Li et al. (2019) achieve scale invariance by sharing weights among kernels with different dilation rates. Their approach can be considered a special case of our proposed B-spline G-CNNs with kernels that do not encode scale interactions. Related work by Tai et al. (2019) and Henriques & Vedaldi (2017) relies on the same Lie group principles as we do in this paper (the Log map) to construct convenient coordinate systems, such as log-polar coordinates Esteves et al. (2018b), to handle equivariance. Such methods are however generally not translation equivariant and do not deal with local symmetries as they act globally on feature maps, much like spatial transformer networks (Jaderberg et al., 2015).

**B-splines and vector fields in deep learning** The current work can be seen as a generalization of the B-spline based $SE(2)$ CNNs of Bekkers et al. (2015a; 2018b), see Sec. 3.3. Closely related is also the work of Fey et al. (2018) in which B-splines are used to generalize CNNs to non-Euclidean

data (graphs). There it is proposed to perform convolution via B-spline kernels on $\mathbb{R}^d$ that take as inputs vectors $\mathbf{u}(i,j) \in \mathbb{R}^d$ that relate any two points $i, j \in \mathcal{G}$ in the graph to each other. How $\mathbf{u}(i,j)$ is constructed is left as a design choice, however, in (Fey et al., 2018) this is typically done by embedding the graph in an Euclidean space where points relate via offset vectors. In our work on Lie G-CNNs, two points $g, g' \in G$ in the Lie group $G$ relate via the logarithmic map $\mathbf{u}(g,g') = \mathrm{Log}\, g^{-1}g'$. Another related approach in which convolutions take place on manifolds in terms of "offset vectors" is the work by Cohen et al. (2019). There, points relate via the exponential map with respect to gauge frames rather than the left-invariant vector fields as in this paper, see App. C.2.

## 3 LIE GROUP CNNs

### 3.1 PRELIMINARIES AND NOTATION

The following describes the essential tools required for deriving a generic framework G-CNNs. Although we treat groups mostly in an abstract setting, we here provide examples for the roto-translation group and refer to App. B for more details, explicit formula's, and figures for several groups. Implementations are available at `https://github.com/ebekkers/gsplinets`.

**Group** A group is defined by a set $G$ together with a binary operator $\cdot$, the group product, that satisfies the following axioms: *Closure*: For all $h, g \in G$ we have $h \cdot g \in G$; *Identiy*: There exists an identity element $e$; *Inverse*: for each $g \in G$ there exists an inverse element $g^{-1} \in G$ such that $g^{-1} \cdot g = g \cdot g^{-1} = e$; and *Associativity*: For each $g, h, i \in G$ we have $(g \cdot h) \cdot i = g \cdot (h \cdot i)$.

**Lie group and Lie algebra** If furthermore the group has the structure of a differential manifold and the group product and inverse are smooth, it is called a *Lie group*. The differentiability of the group induces a notion of infinitesimal generators (see also the exponential map below), which are elements of the Lie algebra $\mathfrak{g}$. The Lie algebra consists of a vector space (of generators), that is typically identified with the tangent space $T_e(G)$ at the identity $e$, together with a bilinear operator called the Lie bracket. In this work the Lie bracket is not of interest and we simply say $\mathfrak{g} = T_e(G)$.

**Exponential and logarithmic map** We expand vectors in $\mathfrak{g}$ in a basis $\{A_i\}_{i=1}^n$ and write $A = \sum_i^n a^i A_i \in \mathfrak{g}$, with components $\mathbf{a} = (a^1, \ldots, a^n) \in \mathbb{R}^n$. This allows us to identify the Lie algebra with $\mathbb{R}^n$. Lie groups come with a logarithmic map that maps elements from the typically non-flat manifolds $G$ to the flat Euclidean vector space $\mathfrak{g}$ via $A = \mathrm{Log}\, g$ (conversely $g = \mathrm{Exp}\, A$), see Fig. 2.

**Semi-direct product groups** In this paper we specifically consider (affine) Lie groups of type $G = \mathbb{R}^d \rtimes H$ that are the semi-direct product of the translation group $\mathbb{R}^d$ with a Lie group $H$ that acts on $\mathbb{R}^d$. Let $h \odot \mathbf{x}$ denote the action of $h \in H$ on $\mathbf{x} \in \mathbb{R}^d$; it describes how a vector in $\mathbb{R}^d$ is transformed by $h$. Then the group product of $\mathbb{R}^d \rtimes H$ is given by

$$g_1 \cdot g_2 = (\mathbf{x}_1, h_1) \cdot (\mathbf{x}_2, h_2) = (\mathbf{x}_1 + h_1 \odot \mathbf{x}_2, h_1 \cdot h_2), \tag{1}$$

with $g_1 = (\mathbf{x}_1, h_1), g_2 = (\mathbf{x}_2, h_2) \in G$, $\mathbf{x}_1, \mathbf{x}_2 \in \mathbb{R}^d$ and $h_1, h_2 \in H$. For example the special Euclidean motion group $SE(2)$ is constructed by choosing $H = SO(2)$, the group of $2 \times 2$ rotation matrices with matrix multiplication as the group product. The group product of $G$ is then given by

$$(\mathbf{x}_1, \mathbf{R}_{\theta_1}) \cdot (\mathbf{x}_2, \mathbf{R}_{\theta_2}) = (\mathbf{x}_1 + \mathbf{R}_{\theta_1}.\mathbf{x_2}, \mathbf{R}_{\theta_1}.\mathbf{R}_{\theta_2}),$$

with $\mathbf{x}_1, \mathbf{x}_2 \in \mathbb{R}^2$ and $\mathbf{R}_{\theta_1}, \mathbf{R}_{\theta_2} \in SO(2)$ rotation matrices parameterized by a rotation angle $\theta_i$, and in which rotations act on vectors in $\mathbb{R}^2$ simply by matrix vector multiplication.

**Group representations** We consider linear transformations $\mathcal{L}_g^{G \to \mathbb{L}_2(X)} : \mathbb{L}_2(X) \to \mathbb{L}_2(X)$ that transform functions (or feature maps) $f \in \mathbb{L}_2(X)$ on some space $X$ as representations of a group $G$ if they share the group structure via

$$(\mathcal{L}_{g_1}^{G \to \mathbb{L}_2(X)} \circ \mathcal{L}_{g_2}^{G \to \mathbb{L}_2(X)} f)(x) = (\mathcal{L}_{g_1 \cdot g_2}^{G \to \mathbb{L}_2(X)} f)(x),$$

with $\circ$ denoting function composition. Thus, a concatenation of two such transformations, parameterized by $g_1$ and $g_2$, can be described by a single transformation parameterized by $g_1 \cdot g_2$. For semi-direct product groups $G = \mathbb{R}^d \rtimes H$ such a representation can be split into

$$\mathcal{L}_g^{G \to \mathbb{L}_2(X)} = \mathcal{L}_{\mathbf{x}}^{\mathbb{R}^d \to \mathbb{L}_2(X)} \circ \mathcal{L}_h^{H \to \mathbb{L}_2(X)}. \tag{2}$$

**Equivariance** An operator $\Phi : \mathbb{L}_2(X) \to \mathbb{L}_2(Y)$ is said to be equivariant to $G$ when it satisfies

$$\forall_{g \in G} : \quad \mathcal{L}_g^{G \to \mathbb{L}_2(Y)} \circ \Phi = \Phi \circ \mathcal{L}_g^{G \to \mathbb{L}_2(X)}. \tag{3}$$

### 3.2 Group convolutional neural networks

Equivariance of artificial neural networks (NNs) with respect to a group $G$ is a desirable property as it guarantees that no information is lost when applying transformations to the input, the information is merely shifted to different locations in the network. It turns out that if we want NNs to be equivariant, then our only option is to use layers whose linear operator is defined by group convolutions. We summarize this in Thm. 1. To get there, we start with the traditional definition of NN layers via

$$\underline{y} = \phi(\mathcal{K}_{\underline{w}}\underline{x} + \underline{b}),$$

with $\underline{x} \in \mathcal{X}$ the input vector, $\mathcal{K}_{\underline{w}} : \mathcal{X} \to \mathcal{Y}$ a linear map parameterized by a weight vector $\underline{w}$, with $\underline{b} \in \mathcal{Y}$ a bias term, and $\phi$ a point-wise non-linearity. In classical neural networks $\mathcal{X} = \mathbb{R}^{N_x}$ and $\mathcal{Y} = \mathbb{R}^{N_y}$ are Euclidean vector spaces and the linear map $\mathcal{K}_{\underline{w}} = \mathbb{R}^{N_y \times N_x}$ is a weight matrix. In this work we instead focus on structured data and consider feature maps on some domain $X$ as functions $\underline{f} : X \to \mathbb{R}^N$, the space of which we denote with $(\mathbb{L}_2(X))^N$. In this case $\mathcal{X} = (\mathbb{L}_2(X))^{N_x}$ and $\mathcal{Y} = (\mathbb{L}_2(Y))^{N_y}$ are the spaces of multi-channel feature maps, $\underline{b} \in \mathbb{R}^{N_y}$, and $\mathcal{K}_{\underline{w}} : \mathcal{X} \to \mathcal{Y}$ is a kernel operator. Now when we constrain the linear operator $\mathcal{K}_{\underline{w}}$ to be equivariant under transformations in some group $G$ we arrive at group convolutional neural networks. This is formalized in the following theorem on equivariant maps between homogeneous spaces (see (Duits & Burgeth, 2007; Kondor & Trivedi, 2018; Cohen et al., 2018a) for related statements).

**Theorem 1.** *Let operator $\mathcal{K} : \mathbb{L}_2(X) \to \mathbb{L}_2(Y)$ be linear and bounded, let $X, Y$ be homogeneous spaces on which Lie group $G$ act transitively, and $\mathrm{d}\mu_X$ a Radon measure on $X$, then*

1. *$\mathcal{K}$ is a kernel operator, i.e., $\exists_{\tilde{k} \in \mathbb{L}_1(Y \times X)} : (\mathcal{K}f)(y) = \int_X \tilde{k}(y, x) f(x) \mathrm{d}\mu_X$,*

2. *under the equivariance constraint of Eq. (3) the map is defined by a one-argument kernel*

$$\tilde{k}(y, x) = \frac{\mathrm{d}\mu_X(g_y^{-1} \odot x)}{\mathrm{d}\mu_X(x)} k(g_y^{-1} \odot x) \tag{4}$$

   *for any $g_y \in G$ such that $y = g_y \odot y_0$ for some fixed origin $y_0 \in Y$,*

3. *if $Y \equiv G/H$ is the quotient of $G$ with $H = \mathrm{Stab}_G(y_0) = \{g \in G | g \odot y_0 = y_0\}$ then the kernel is constrained via*

$$\forall_{h \in H}, \forall_{x \in X} : \quad k(x) = \frac{\mathrm{d}\mu_X(g_y^{-1} \odot x)}{\mathrm{d}\mu_X(x)} k(h^{-1} \odot x), \tag{5}$$

*Proof.* See App. A $\hfill\square$

**Corollary 1.** *If $X = \mathbb{R}^d$ is a homogeneous space of an affine Lie group $G = \mathbb{R}^d \rtimes H$ and $\mathrm{d}\mu_X(x) = \mathrm{d}x$ is the Lebesgue measure on $\mathbb{R}^d$ then the kernel front-factor simplifies to $\frac{\mathrm{d}\mu_X(g^{-1} \odot x)}{\mathrm{d}\mu_X(x)} = \frac{1}{|\det h|}$ with $|\det h|$ denoting the determinant of the matrix representation of $h$, for any $g = (\mathbf{x}, h) \in G$. If $X = G$ and $\mathrm{d}\mu_X(x)$ is a Haar measure on $G$ then $\frac{\mathrm{d}\mu_X(g^{-1} \odot x)}{\mathrm{d}\mu_X(x)} = 1$.*

In view of Thm. 1 we see that standard CNNs are a special case of G-CNNs that are equivariant to translations. In this case the domain of the feature maps $X = \mathbb{R}^d$ coincides with the space of translation vectors in the translation group $G = (\mathbb{R}^d, +)$. It is well known that if we want the networks to be translation *and* rotation equivariant ($G = \mathbb{R}^d \rtimes SO(d)$), but stick to planar feature maps, then the kernels should be rotation invariant, which of course limits representation power. This constraint is due Thm. 1 item 3, since the domain of such features maps is a quotient group $\mathbb{R}^d \equiv G/\{\mathbf{0}\} \times SO(d)$ in which rotations ($SO(d)$) are factored out of the roto-translation group $G = \mathbb{R}^d \rtimes SO(d)$. Thus, in order to maximize representation power (without constrains on $\tilde{k}$) the feature maps should be lifted to the higher dimensional domain of the group itself (i.e. $Y = G$). We therefore propose to build G-CNNs with the following 3 types of layers (illustrated in Fig. 1):

- **Lifting layer** ($X = \mathbb{R}^d, Y = G$)**:** In this layer $\mathcal{K}$ is defined by lifting correlations

$$(\mathcal{K}f)(g) = (k \tilde{\star} f)(g) := \frac{1}{|\det h|} \left( \mathcal{L}_g^{G \to \mathbb{L}_2(\mathbb{R}^d)} k, f \right)_{\mathbb{L}_2(\mathbb{R}^d, \mathrm{d}\mathbf{x})},$$

   with $g = (\mathbf{x}, h)$, which by splitting of the representation (Eq. (2)) can be written as

$$\boxed{(\mathcal{K}f)(g) = (k \tilde{\star} f)(g) = (k_h \star_{\mathbb{R}^d} f)(\mathbf{x}),} \tag{6}$$

with $k_h(\mathbf{x}) = \frac{1}{|\det h|} \left( \mathcal{L}_h^{H \to \mathbb{L}_2(\mathbb{R}^d)} k \right)(\mathbf{x})$ the transformed kernel. Lifting correlations thus match a kernel with the input feature map under all possible transformations in $G$.

- **Group correlation layer** ($X = G, Y = G$)**:** In this case $\mathcal{K}$ is defined by group correlations

$$(\mathcal{K}F)(g) = (K \star F)(g) := \left( \mathcal{L}_g^{G \to \mathbb{L}_2(G)} K, F \right)_{\mathbb{L}_2(G, \mathrm{d}\mu)} = \int_G K(g^{-1}\tilde{g}) F(\tilde{g}) \mathrm{d}\mu(\tilde{g}),$$

with $\mathrm{d}\mu(g)$ a Haar measure on $G$. We can again split this cross-correlation into a transformation of $K$ followed by a spatial cross-correlation via

$$\boxed{(\mathcal{K}F)(g) = (K \star F)(g) = (K_h \star_{\mathbb{R}^d} F)(\mathbf{x}),} \tag{7}$$

with $K_h(\tilde{\mathbf{x}}, \tilde{h}) = K(h^{-1} \odot \tilde{\mathbf{x}}, h^{-1} \cdot \tilde{h})$ the convolution kernel transformed by $h \in H$ and in which we overload $\star_{\mathbb{R}^d}$ to indicate cross-correlation on the $\mathbb{R}^d$ part of $G = \mathbb{R}^d \rtimes H$.

- **Projection layer** ($X = G, Y = \mathbb{R}^d$)**:** In this case $\mathcal{K}$ is a linear projection defined by

$$\boxed{(\mathcal{K}F)(\mathbf{x}) = \int_H F(\mathbf{x}, \tilde{h}) \mathrm{d}\mu(\tilde{h}),} \tag{8}$$

where we simply integrate over $H$ instead of using a kernel that would otherwise be constant over $H$ and spatially isotropic with respect to $H$.

App. B provides explicit formula's for these layers for several groups. E.g., in the $SE(2)$ we get:

$SE(2)$-lifting: $\qquad (k\tilde{\star}f)(\mathbf{x}, \theta) = \int_{\mathbb{R}^2} k(\mathbf{R}_\theta^{-1}.(\mathbf{x}' - \mathbf{x})) f(\mathbf{x}') \mathrm{d}\mathbf{x}',$

$SE(2)$-correlation: $(K\star F)(\mathbf{x}, \theta) = \int_{\mathbb{R}^2} \int_{S^1} K(\mathbf{R}_\theta^{-1}.(\mathbf{x}' - \mathbf{x}), \theta' - \theta \bmod 2\pi) F(\mathbf{x}', \theta') \mathrm{d}\mathbf{x}' \mathrm{d}\theta'.$

### 3.3 B-Splines on Lie groups

Central in our formulation of G-CNNs is the transformation of convolution kernels under the action of $H$ as described above in Eqs. (6) and (7) in the continuous setting. For the implementation of G-CNNs the kernels and their transformations need to be sampled on a discrete grid. We expand on the idea's in (Bekkers et al., 2015a; 2018b; Weiler et al., 2018b) to express the kernels in an analytic form which we can then sample under arbitrary transformations in $G$ to perform the actual computations. In particular we generalize the approach of Bekkers et al. (2015a; 2018b) to expand group correlation kernels in a basis of shifted cardinal B-splines, which are localized polynomial functions on $\mathbb{R}^d$ with finite support. In (Bekkers et al., 2015a; 2018b), B-splines on $\mathbb{R}^d$ could be used to construct kernels on $SE(2)$ by identifying the group with the space of positions and orientations and simply using periodic splines on the 1D orientation axis $S^1$. However, in order to construct B-splines on arbitrary Lie groups, we need a generalization. In the following we propose a new definition of B-splines on Lie groups $H$ which enables us to construct the kernels on $\mathbb{R}^d \rtimes H$ that are required in the G-correlations (Eq. (7)).

**Definition 1** (Cardinal B-spline on $\mathbb{R}^n$). *The 1D cardinal B-Spline of degree $n$ be is defined as*

$$B^n(x) := \left( 1_{\left[ -\frac{1}{2}, \frac{1}{2} \right]} *^{(n)} 1_{\left[ -\frac{1}{2}, \frac{1}{2} \right]} \right)(x), \tag{9}$$

*where $*^{(n)}$ denotes $n$-fold convolution of the indicator function $1_{\left[ -\frac{1}{2}, \frac{1}{2} \right]}$. The multi-variate cardinal B-spline on $\mathbb{R}^d$, with coordinates $\mathbf{x} = (x_0, \ldots, x_d)^T \in \mathbb{R}^d$, is defined via the tensor product*

$$B^{\mathbb{R}^d, n}(\mathbf{x}) := \underbrace{(B^n \otimes \cdots \otimes B^n)(\mathbf{x})}_{d \ \text{times}} = B^n(x_0) B^n(x_1) \ldots B^n(x_d). \tag{10}$$

Cardinal B-splines are piece-wise polynomials and are localized on support $[-\frac{n+1}{2}, \frac{n+1}{2}]$. Functions can be expanded in a basis of shifted cardinal B-splines, which we simply refer to as B-splines.

**Definition 2** (B-splines on $\mathbb{R}^n$). *A B-spline is a function $f : \mathbb{R}^d \to \mathbb{R}$ expanded in a basis that consists of shifted and scaled copies of the cardinal B-spline*

$$f(\mathbf{x}) := \sum_{i=1}^{N} c_i B^{\mathbb{R}^d, n}\left(\frac{\mathbf{x} - \mathbf{x}_i}{s_{\mathbf{x}}}\right), \tag{11}$$

*and is fully characterized by spline degree $n$, scale $s_{\mathbf{x}}$, set of centers $\{\mathbf{x}_i\}_{i=1}^N$ with $\mathbf{x}_i \in \mathbb{R}^d$ and corresponding coefficients $\mathbf{c} = (c_1, c_2, \ldots, c_N)^T \in \mathbb{R}^N$. The B-spline is called uniform if the set of centers $\{\mathbf{x}_i\}_{i=1}^N$ forms a uniform grid on $\mathbb{R}^d$, in which the distance $\|\mathbf{x}_j - \mathbf{x}_i\|$ between neighbouring centers $\mathbf{x}_i, \mathbf{x}_j \in \mathbb{R}^d$ is constant along each axis and equal to $s_{\mathbf{x}}$.*

**Definition 3** (B-splines on Lie group $H$). *A B-spline on $H$ is a function $f : H \to \mathbb{R}$ expanded in a basis that consists of shifted (by left multiplication) and scaled copies of the cardinal B-spline*

$$f(h) := \sum_{i=1}^{N} c_i B^{\mathbb{R}^d, n}\left(\frac{\mathrm{Log}\, h_i^{-1} h}{s_h}\right), \tag{12}$$

*with $h \in H$ and $\mathrm{Log} : H \to \mathfrak{h}$ the logarithmic map on $H$. The B-spline is fully characterized by the spline degree $n$, scale $s_h$, set of centers $\{h_i\}_{i=1}^N$ with $h_i \in H$ and corresponding coefficients $\mathbf{c} = (c_1, c_2, \ldots, c_N)^T \in \mathbb{R}^N$. The spline is called uniform if the distance $\|\mathrm{Log}\, h_i^{-1} h_j\|$ between neighbouring centers $h_i, h_j \in H$ is constant.*

Examples of B-splines on Lie groups $H$ are given in Fig. 3. In this paper we choose to expand convolution kernels on $G = \mathbb{R}^d \rtimes H$ as the tensor product of B-splines on $\mathbb{R}^d$ and $H$ respectively and obtain functions $f : \mathbb{R}^d \times H$ via

$$f(\mathbf{x}, h) := \sum_{i=1}^{N} c_i B^{\mathbb{R}^d, n}\left(\frac{\mathbf{x} - \mathbf{x}_i}{s_{\mathbf{x}}}\right) B^{\mathbb{R}^d, n}\left(\frac{\mathrm{Log}\, h_i^{-1} h}{s_h}\right). \tag{13}$$

Note that that one could also directly define B-splines on $G$ via (12), however, this splitting ensures we can use a regular Cartesion grid on the $\mathbb{R}^d$ part. In our experiments we use B-splines as in (13) and consider the coefficients $\mathbf{c}$ as trainable parameters and the centers ($\mathbf{x}_i$ and/or $\mathbf{h_i}$) and scales ($s_{\mathbf{x}}$ and/or $s_h$) are fixed by design. Some design choices are the following (and illustrated in Fig. 4).

**Global vs localized uniform B-splines** The notion of a *uniform* B-spline globally covering $H$ exists only for a small set of Lie groups, e.g. for any 1D group and abelian groups, and it is not possible to construct uniform B-splines on Lie groups in general due to non-zero commutators. Nevertheless, we find that it is possible to construct *approximately uniform* B-splines on compact groups either by constructing a grid of centers $\{h_i\}_{i=1}^N$ on $H$ that approximately uniformly covers $H$, e.g. by using a repulsion model in which $\|\mathrm{Log}\, h_i^{-1} \cdot h_j\|$ between any two grid points $h_i, h_j \in H$ is maximized (as is done in Fig. 3), or by specifying a uniform localized grid on the lie algebra $\mathfrak{h}$ and obtaining the centers via the exponential map. The latter approach is in fact possible for any Lie group and leads to a notion of localized convolution kernels that have a finite support on $H$, see Fig. 4.

**Atrous B-splines** Atrous convolutions, i.e. convolutions with sparse kernels defined by weights interleaved with zeros (Holschneider et al., 1990), are commonly used to increase the effective receptive field size and add a notion of scale to deep CNNs (Yu & Koltun, 2016; Chen et al., 2018). Atrous convolution kernels can be constructed with B-splines by fixing the scale factors $s_{\mathbf{x}}$ and $s_h$, e.g. to the grid size, and increasing the distance between the center points $\mathbf{x}_i$ and $h_i$.

**Non-uniform/deformable B-splines** In non-uniform B-splines the centers $\mathbf{x}_i$ and $h_i$ do not necessarily need to lie on a regular grid. Then, deformable CNNs, first proposed by Dai et al. (2017), are obtained by treating the centers as trainable parameters. For B-spline CNNs on $\mathbb{R}^d$ of order $n = 1$ this in fact leads to the deformable convolution layers as defined in (Dai et al., 2017).

**Modular design** The design of G-correlation layers (Eqs. (6-8)) using B-spline kernels (Eqs. (11-13)) results in a generic and modular construction of G-CNNs that are equivariant to Lie groups $G$ and only requires a few group specific definitions (see examples in App. B): The group structure of $H$ (group product $\cdot$ and inverse), the action $\odot$ of $H$ on $\mathbb{R}^d$ (together with the group structure of $H$ this automatically defines the structure of $G = \mathbb{R}^d \rtimes H$), and the logarithmic map $\mathrm{Log} : H \to \mathfrak{h}$.

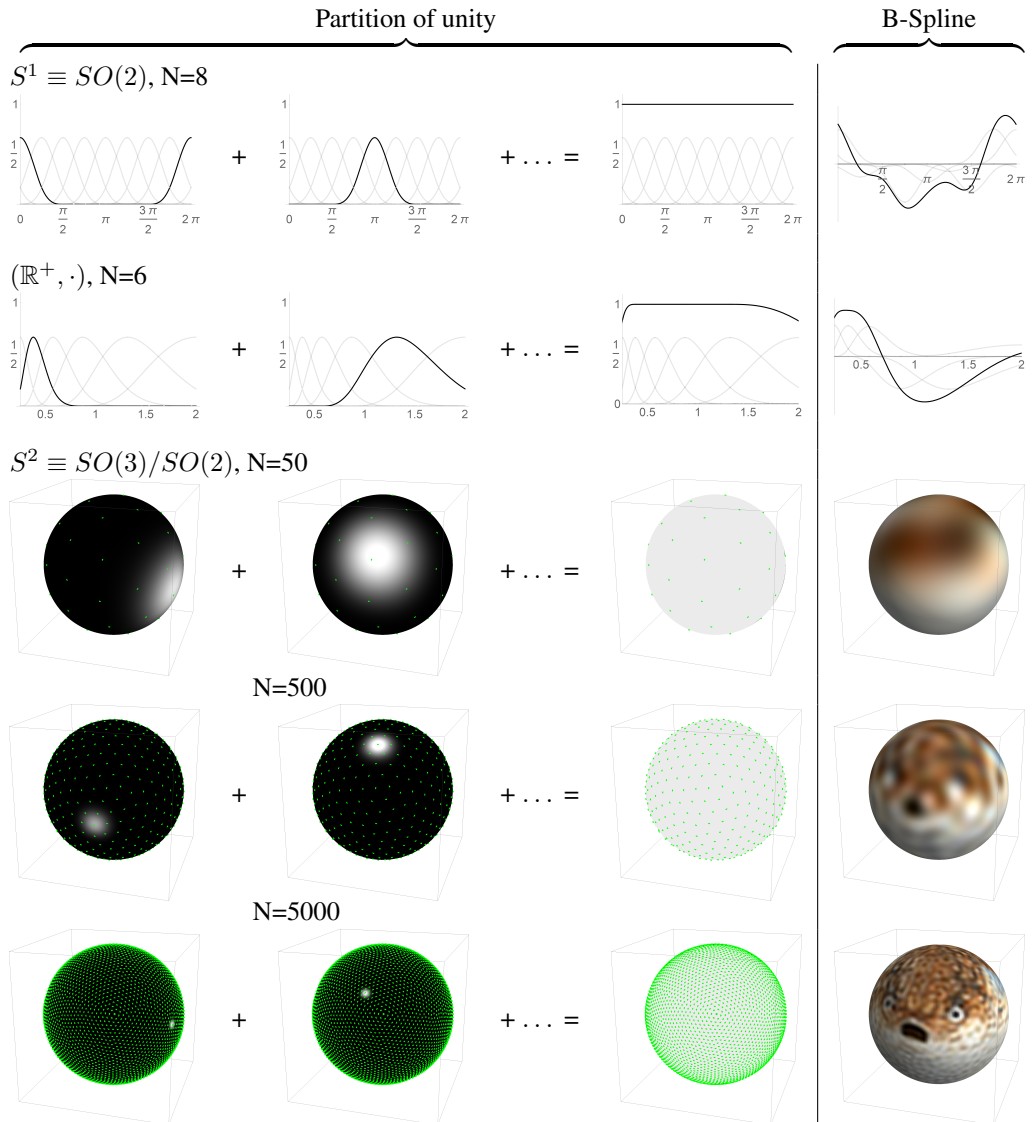

Figure 3: *Left*: The sum of all B-spline basis functions add up to one, illustrating partition of unity on the 2D rotation group $SO(2)$ (row 1), scaling/dilation group $(\mathbb{R}^+, \cdot)$ (row 2), and the sphere $S^2$ treated as the quotient group $SO(3)/SO(2)$, with B-spline centers indicated with green dots (row 3-5). *Right*: A random B-Spline on $SO(2)$ (row 1) and $(\mathbb{R}^+, \cdot)$ (row 2) and reconstruction of a color texture on the sphere $S^2$ at several scales (row 3-5) to illustrate multi-scale properties.

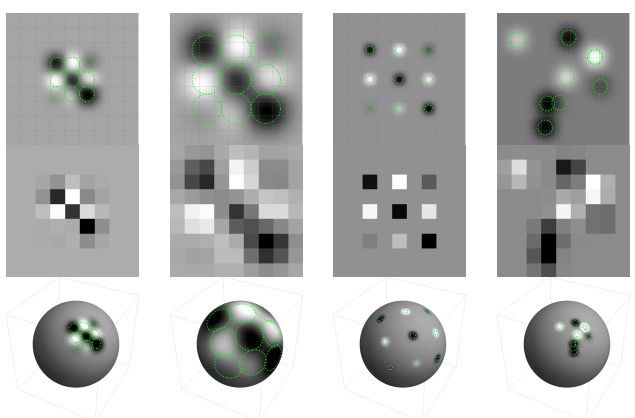

Figure 4: A B-Spline on $\mathbb{R}^2$ (row 1), sampled on a grid (row 2), and a B-spline on the sphere (row 3). From left to right: a localized kernel, scaled kernel by increasing $s_\mathbf{x}$ and $s_h$, atrous kernel, deformable kernel. A green circle is drawn around each B-spline center with radius $\frac{1}{2}s_\mathbf{x}$ or $\frac{1}{2}s_h$ to indicate the individual basis functions.

## 4 EXPERIMENTS

### 4.1 ROTO-TRANSLATION CNNS

**Data** The PatchCamelyon (PCam) dataset (Veeling et al., 2018) consists of 327,680 RGB patches taken from histopathologic scans of lymph node sections and is derived from Camelyon16 (Ehteshami Bejnordi et al., 2017). The patches are binary labeled for the presence of metastasis. The classification problem is truly rotation invariant as image features appear under arbitrary rotations at all levels of abstraction, e.g. from edges (low-level) to individual cells to the tissue (high-level).

**Experiments** G-CNNs ensure roto-translation equivariance both locally (low-level) and globally (high-level) and invariance is achieved by means of pooling. In our experiments we test the performance of roto-translation G-CNNs (with $G = \mathbb{R}^2 \rtimes SO(2)$) against a 2D baseline and investigate the effect of different choices (local, global, atrous) for defining the kernels on the $SO(2)$-part of the network, cf. Eq. (13) and Fig. 4. Each network has the same architecture (detailed in App. D) but the kernels are sampled with varying resolution on $H = SO(2)$, denoted with $N_h$, and with varying resolution of the B-splines, which is achieved by varying $s_h$ and the number of basis functions on $H$, denoted with $N_k$. Each network has approximately the same number of trainable weights. Each network is trained with data-augmentation (see App. D for details), but for reference we also compare to a 2D and a $SE(2)$ model ($N_k = N_h = 12$) which are trained without $90°$ rotation augmentation.

The results are summarized in Fig. 5. Here $N_h = 1$ means the kernels are transformed for only one rotation, which coincides with standard 2D convolutions (our baseline). A result labeled "dense" with $N_h = 16$ and $N_k = 8$ means the convolution kernels are rotated 16 times and the kernels are expanded in a B-spline basis with 8 basis functions to fully cover $H$. The label "local" means the basis is localized with $N_k$ basis functions with a spacing of $s_h = \frac{2\pi}{16}$ between them, with $s_h$ equal to the grid resolution. Atrous kernels are spaced equidistantly on $H$ and have $s_h \ll \frac{2\pi}{N_h}$.

**Results** We generally observe that a finer sampling of $SO(2)$ leads to better results up until $N_h = 12$ after which results slightly degrade. This is line with findings in (Bekkers et al., 2018a). The degradation after this point could be explained by overfitting; there is a limit on the resolution of the signal generated by rotating 5x5 convolution kernels; at some point the splines are described in more detail than the data and thus an unnecessary amount of coefficients are trained. One could still benefit from sampling coarse kernels (low $N_k$) on a fine grid (high $N_h$), e.g. compare the cases $N_h > N_k$ for fixed $N_k$. This is in line with findings in (Weiler et al., 2018b) where a fixed circular harmonic basis is used. Generally, atrous kernels tend to outperform dense kernels as do the localized kernels in the low $N_k$ regime. Finally, comparing the models with and without $90°$ augmentation show that such augmentations are crucial for the 2D model but hardly affect the $SE(2)$ model. Moreover, the $SE(2)$ model *without* outperforms the 2D model *with* augmentation. This confirms the theory: G-CNNs guarantee both local and global equivariance by construction, whereas with augmentations valuable network capacity is spend on learning (only) global invariance. The very modest drop in the $SE(2)$ case may be due to the discretization of the network on a grid after which it is no longer purely equivariant but rather approximately, which may be compensated for via augmentations.

### 4.2 SCALE-TRANSLATION CNNS

**Data** The CelebA dataset (Liu et al., 2015) contains 202,599 RGB images of varying size of celebrities together with labels for attributes (hair color, glasses, hat, etc) and 5 annotated facial landmarks (2 eyes, 1 nose, 2 corners of the mouth). We reformatted the data as follows. All images are isotropically scaled to a maximum width or height of 128 and if necessary padded in the other dimension with zeros to obtain a size of 128x128. For each image we took the distance between the eyes as a reference for the size of the face and categorized each image into above and below average size. For each unique celebrity with at least 1 image per class, we randomly sampled 1 image per class. The final dataset consists of 17,548 images of 128x128 of 8,774 celebrities with faces at varying scales. Each image is labeled with 5 heatmaps constructed by sampling a Gaussian with standard deviation 1.5 centered around each landmark.

**Experiments** We train a scale-translation G-CNN (with $G = \mathbb{R}^2 \rtimes \mathbb{R}^+$) with different choices for kernels. The "dense" networks have kernels defined over the whole discretization of $H = \mathbb{R}^+$ and thus consider interactions between features at all scales. The "local" networks consider only

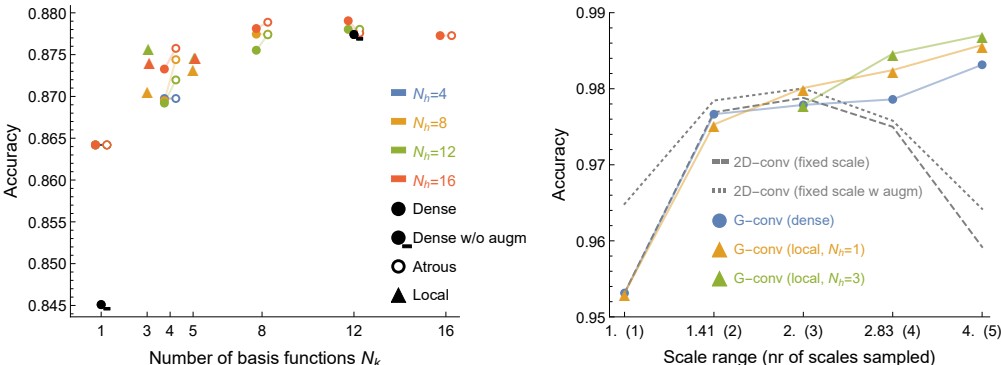

Figure 5: Left: results of roto-translation G-CNNs on tumor classification (PCam dataset). Right: results of scale-translation G-CNNs on landmark localization (CelebA dataset).

interaction between neighbouring scales via localized kernels ($N_k = 3$) or no scale interaction at all ($N_k = 1$). Either way, each G-CNN is a multi-scale network in which kernels are applied at a range of scales. We compared against a 2D baseline with fixed-scale kernels which we tested for several scales separately. In the G-CNNs, $H$ is uniformly sampled (w.r.t. to the metric on $H$) on a fixed scale range, generating the discrete sets $H_d = \{e^{(i-1)s_h}\}_{i=1}^{N_h}$ with $s_h = \frac{1}{2}\ln 2$. Each G-CNN is sampled with the same resolution in $H$ with $s_h$, and each B-spline basis function is centered on the discrete grid (i.e. $h_i \in H_d$). We note that the discretization of $H$ is formally no longer a group as it is not closed, however, the group structure still applies locally. The result is that information may leak out of the domain in a similar way as happens spatially in standard zero-padded 2D CNNs (translational G-CNNs), in which the discretized domain of translations is also no longer (locally) compact. This information leak can be avoided by using localized kernels of size $N_k = 1$ along the $H$ axis, as is also done in (Worrall & Welling, 2019; Li et al., 2019). The networks are trained without data-augmentation, except for the 2D network, which for reference we train with and without random scale augmentations at train-time. Augmentations were done with a random scale-factor between 1 and 1.4. We found that scale augmentations beyond 1.4 did not improve results.

**Results** Fig. 5 summarizes the results. By testing our 2D baseline at several scales we observe that there is an optimal scale ($h = 2$) that gives a best trade off between the scale variations in the data. This set of experiments is also used to rule out the idea that G-CNNs outperform the 2D baseline simply because they have a larger effective receptive field size. For large scale ranges the G-CNNs start to outperform 2D CNNs as these networks consider both small and large scale features simultaneously (multi-scale behavior). Comparing different G-CNN kernel specifications we observe that enabling interactions between neighbouring scales, via localized kernels on $H$ ("local", $N_h = 3$), outperforms both all-scale interactions ("dense") and no-scale interaction at all ($N_h = 1$). This finding is in line with those in (Worrall & Welling, 2019). Finally, although 2D CNNs moderately benefit from scale augmentations, it is not possible to achieve the performance of G-CNNs. It seems that a multi-scale approach (G-CNNs) is essential.

## 5 CONCLUSION

This paper presents a flexible framework for building G-CNNs for arbitrary Lie groups. The proposed B-spline basis functions, which are used to represent convolution kernels, have unique properties that cannot be achieved by classical Fourier based basis functions. Such properties include the construction of localized, atrous, and deformable convolution kernels. We experimentally demonstrated the added value of localized and atrous group convolutions on two different applications, considering two different groups. In particular in experiments with scale-translation G-CNNs, kernel localization was important. The B-spline basis functions can be considered as smooth pixels on Lie groups and they enable us to design G-CNNs using familiar notions from classical CNN design (localized, atrous, and deformable convolutions). Future work will focus on exploring these options further in new applications that could benefit from equivariance constraints, for which the tools now are available for a large class of transformation groups via the proposed Lie group B-splines.

ACKNOWLEDGMENTS

Remco Duits (Eindhoven University of Technology (TU/e)) is gratefully acknowledged for his contributions to the formulation and proof of Thm. 1 and for helpful discussions on Lie groups. I thank Maxime Lafarge (TU/e) for advice on setting up the PCam experiments. This work is part of the research programme VENI with project number 17290, which is (partly) financed by the Dutch Research Council (NWO).

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

## A    PROOF OF THEOREM 1

The following proofs the three sub-items of Thm. 1.

1. It follows from Dunford-Pettis Theorem, see e.g. (Arendt & Bukhvalov, 1994, Thm 1.3), (Kantorovich & Akilov, 1982, Ch 9, Thm 5), or (Duits, 2005, Thm 1), that if $\mathcal{K}$ is linear and bounded it is an integral operator.

2. The left-equivariance constraint then imposes bi-left-invariance of the kernel $\tilde{k}$ as follows, where $\forall_{g \in G}$ and $\forall_{f \in \mathbb{L}_2(X)}$:

$$(\mathcal{K} \circ \mathcal{L}_g^{G \to \mathbb{L}_2(X)})(f) = (\mathcal{L}_g^{G \to \mathbb{L}_2(Y)} \circ \mathcal{K})(f) \quad \Leftrightarrow$$

$$\int_X \tilde{k}(y, x) f(g^{-1} x) \mathrm{d}x = \int_X \tilde{k}(g^{-1} y, x) f(x) \mathrm{d}x \overset{\text{in r.h.s. integral } x \leftarrow g^{-1} x}{\Leftrightarrow}$$

$$\int_X \tilde{k}(y, x) f(g^{-1} x) \mathrm{d}x = \int_X \tilde{k}(g^{-1} y, g^{-1} x) f(g^{-1} x) \mathrm{d}(g^{-1} x) \Leftrightarrow$$

$$\int_X \tilde{k}(y, x) f(g^{-1} x) \mathrm{d}x = \int_X \tilde{k}(g^{-1} y, g^{-1} x) f(g^{-1} x) \tfrac{1}{|\det g|} \mathrm{d}x. \tag{14}$$

Since (14) should hold for all $f \in \mathbb{L}_2(X)$ we obtain

$$\forall_{g \in G} : \qquad \tilde{k}(y, x) = \tfrac{1}{|\det g|} \tilde{k}(g^{-1} y, g^{-1} x). \tag{15}$$

Furthermore, since $G$ acts transitively on $Y$ we have that $\forall_{y, y_0 \in Y} \exists_{g_y \in G}$ such that $y = g_y y_0$ and thus

$$\tilde{k}(y, x) = \tilde{k}(g_y\, y_0, x) \overset{(15)}{=} \tfrac{1}{|\det g_y|} \tilde{k}(y_0, g_y^{-1} x) =: \tfrac{1}{|\det g_y|} k(g_y^{-1} x)$$

for every $g_y \in G$ such that $y = g_y\, y_0$ with arbitrary fixed origin $y_0 \in Y$.

3. Every homogeneous space $Y$ of $G$ can be identified with a quotient group $G/H$. Choose an origin $y_0 \in Y$ s.t. $\forall_{h \in H} : h\, y_0 = y_0$, i.e., $H = \mathrm{Stab}_G\, y_0$, then

$$\tilde{k}(y_0, x) = \tilde{k}(h\, y_0, x) \Leftrightarrow k(x) = \tfrac{1}{|\det h|} k(h^{-1} x).$$

We further remark that When $Y \equiv G = G/\{e\}$, with $e \in G$ the identify element of $G$, the symmetry constraint of Eq. (5) vanishes. Thus, in order to construct equivariant maps without constraints on the kernel the functions should be lifted to the group $G$.

## B    EXAMPLES OF LIE GROUPS

In the following sub-sections some explicit examples of Lie groups $H$ are given, together with their actions on $\mathbb{R}^d$ and the Log operators. The required tools for building B-spline based G-CNNs for Lie groups of the form $G = \mathbb{R}^d \rtimes H$ are then automatically derived from these core definitions. E.g., the action $\odot$ of $H$ on a space $X$ defines a left-regular representation on functions on $X$ via

$$(\mathcal{L}_g^{H \to \mathbb{L}_2(X)} f)(x) = f(g^{-1} \odot x).$$

When $X = G$ is the group itself, the action equals the group product. The group structure of semi-direct product groups $G = \mathbb{R}^d \rtimes H$ is automatically derived from the action of $H$ on $\mathbb{R}^d$, see Eq. (1) and is in turn used to define the representations (see Eq. (2)). Some examples are given below.

### B.1    TRANSLATION GROUP $G = (\mathbb{R}^d, +)$

The group of translations is given by the space of translation vectors $\mathbb{R}^d$ with the group product and inverse given by

$$g \cdot g' = (\mathbf{x} + \mathbf{x}')$$
$$g^{-1} = (-\mathbf{x}),$$

with $g = (\mathbf{x}), g' = (\mathbf{x}') \in G$ with $\mathbf{x}, \mathbf{x}' \in \mathbb{R}^d$. The identity element is $e = (\mathbf{0})$. The left-regular representation on $d$-dimensional functions $f \in \mathbb{L}_2(\mathbb{R}^d)$ produces translations of $f$ via

$$(\mathcal{L}_g^{G \to \mathbb{L}_2(\mathbb{R}^d)} f)(g') = f(g^{-1} \cdot g') = f(\mathbf{x}' - \mathbf{x}).$$

Now, since the space of translations can be identified with $\mathbb{R}^d$, the lifting (Eq. (6)) and group correlations (Eq. (7)) coincide with the standard definition of cross-correlation. I.e.,

$$\boxed{(k \tilde{\star} f)(\mathbf{x}) = (k \star f)(\mathbf{x}) = \int_{\mathbb{R}^2} k(\mathbf{x}' - \mathbf{x}) f(\mathbf{x}') d\mathbf{x}'.}$$

The logarithmic map is simply given by

$$\mathrm{Log}\, g = \mathbf{x},$$

so the convolution kernels can be defined by regular splines on $\mathbb{R}^d$, cf. Eq. (11).

## B.2 The 2D rotation group $H = SO(2)$

The special orthogonal group $SO(2)$ consists of all orthogonal $2 \times 2$ matrix with determinant 1, i.e., rotation matrices of the form

$$\mathbf{R}_\theta = \begin{pmatrix} \cos\theta & -\sin\theta \\ \sin\theta & \cos\theta \end{pmatrix},$$

and the group product and inverse is given by the matrix product and matrix inverse:

$$h \cdot h' = (\mathbf{R}_\theta . \mathbf{R}_{\theta'}) = (\mathbf{R}_{\theta + \theta'})$$
$$h^{-1} = (\mathbf{R}_\theta^{-1}),$$

with $h = (\mathbf{R}_\theta), h' = (\mathbf{R}_{\theta'}) \in SO(2)$ with $\theta, \theta' \in S^1$. The identity element is $e = (\mathbf{R}_0) = (\mathbf{I})$. The action of $H$ on $\mathbb{R}^2$ is given by matrix vector multiplication:

$$h \odot \mathbf{x} = \mathbf{R}_\theta . \mathbf{x},$$

with $\mathbf{x} \in \mathbb{R}^2$.

Combining the group structure of 2D translations with rotations in $SO(2)$ as a semi-direct product group gives us the roto-translation group $SE(2) = \mathbb{R}^2 \rtimes SO(2)$, also known as the special Euclidean motion group. The group structure of $SE(2)$ is automatically derived from that of $SO(2)$, see Sec. 3.1 for details.

The left-regular representation of $SO(2)$ are given by

$$(\mathcal{L}_h^{SO(2) \to \mathbb{L}_2(\mathbb{R}^2)} f)(\mathbf{x}') = f(h^{-1} \odot \mathbf{x}') = f(\mathbf{R}_\theta^{-1} . \mathbf{x}'),$$
$$(\mathcal{L}_h^{SO(2) \to \mathbb{L}_2(SO(2))} F)(h') = F(h^{-1} \cdot h') = F(\mathbf{R}_{\theta' - \theta}),$$
$$(\mathcal{L}_h^{SO(2) \to \mathbb{L}_2(SE(2))} F)(\mathbf{x}', h') = F(h^{-1} \odot \mathbf{x}', h^{-1} \cdot h') = F(\mathbf{R}_\theta^{-1} . \mathbf{x}', \mathbf{R}_{\theta' - \theta}).$$

Note that the latter two representations in terms of the rotation parameters $\theta, \theta' \in S^1$ represents the periodic shift $\theta' - \theta \mod 2\pi$ along the rotation axis. See Fig. 6 for an illustration for the transformation of $SE(2)$ convolution kernels via the representation $\mathcal{L}_h^{SO(2) \to \mathbb{L}_2(SE(2))}$. The determinant of the Jacobian of the action of $H$ on $\mathbb{R}^d$, see corollary 1, is $|\det h| = 1$. Using the above group structure we can write out the explicit forms for the lifting (Eq. (6)) and group correlations (Eq. (7)) as follows:

$SE(2)$-lifting: $\boxed{(k \tilde{\star} f)(\mathbf{x}, \theta) = \int_{\mathbb{R}^2} k(\mathbf{R}_\theta^{-1} . (\mathbf{x}' - \mathbf{x})) f(\mathbf{x}') d\mathbf{x}',}$

$SE(2)$-correlation: $\boxed{(K \star F)(\mathbf{x}, \theta) = \int_{\mathbb{R}^2} \int_{S^1} K(\mathbf{R}_\theta^{-1} . (\mathbf{x}' - \mathbf{x}), \theta' - \theta \mod 2\pi) F(\mathbf{x}', \theta') d\mathbf{x}' d\theta'.}$

The logarithmic map on $H$ is given by the matrix logarithm

$$\mathrm{Log}\, \mathbf{R}_\theta = \begin{pmatrix} 0 & -\theta \mod 2\pi \\ \theta \mod 2\pi & 0 \end{pmatrix} \in T_e(SO(2)) = \mathrm{span}\{A_1\}$$

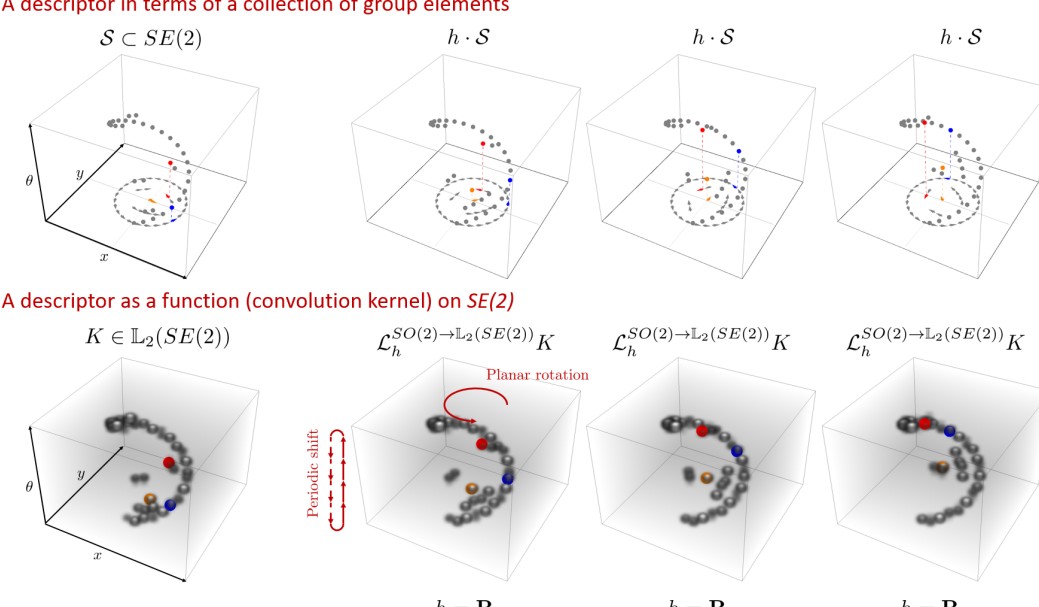

Figure 6: **Top row**: In group theoretical terms we can describe a smiley face via a collection of group elements $\mathcal{S} \subset SE(2)$, which e.g. represents the locations and orientations of low-level features such as edges/lines. Such a collection of points transforms via left multiplication, e.g. $h \cdot \mathcal{S} := \{(\mathbf{0}, h) \cdot g \mid g \in \mathcal{S}\}$, with $h \in SO(2)$. **Bottom row**: In a group convolutional setting we can work with convolution kernels which are functions on $SE(2)$ that assign weights to locations in which locally oriented features are expected. Such kernels transform via the group representation.

Which in terms of the Lie algebra basis $\{A_1\}$ with $A_1 = \begin{pmatrix} 0 & -1 \\ 1 & 0 \end{pmatrix}$ gives a vector with coefficient $a^1 = \theta \bmod 2\pi$. The B-spline basis, centered around each $h_i = (\mathbf{R}_{\theta_i}) \in H$ with scale $s_h \in \mathbb{R}$, as depicted in Fig. 3, is thus computed via

$$B^{\mathbb{R},n}\left(\frac{\operatorname{Log} h_i^{-1} \cdot h}{s_h}\right) = B^{\mathbb{R},n}\left(\frac{\theta - \theta_i \bmod 2\pi}{s_h}\right).$$

## B.3 SCALING GROUP $H = (\mathbb{R}^+, \times)$

We call the positive real line $\mathbb{R}^+$, together with multiplication, the scaling group. The group product and inverse are given by

$$h \cdot h' = (ss')$$
$$h^{-1} = \left(\tfrac{1}{s}\right),$$

with $h = (s), h' = (s') \in H$ with $s, s' \in \mathbb{R}^+$. The identity element is $e = (1)$. The action of $H$ on $\mathbb{R}^d$ is given by scalar multiplication

$$h \odot \mathbf{x} = s\,\mathbf{x},$$

with $\mathbf{x} \in \mathbb{R}^d$. By combining 2D translations with scaling as a semi-direct product group we obtain the scale-translation group, which we denote with $\mathbb{R}^2 \rtimes \mathbb{R}^+$. The left-regular representation of the scaling group are given by

$$(\mathcal{L}_h^{\mathbb{R}^+ \to \mathbb{L}_2(\mathbb{R}^d)} f)(\mathbf{x}') = f(h^{-1} \odot \mathbf{x}') = f(\tfrac{1}{s}\mathbf{x}'),$$
$$(\mathcal{L}_h^{\mathbb{R}^+ \to \mathbb{L}_2(\mathbb{R}^+)} F)(h') = F(h^{-1} \cdot h') = F(\tfrac{s'}{s}),$$
$$(\mathcal{L}_h^{\mathbb{R}^+ \to \mathbb{L}_2(\mathbb{R}^2 \times \mathbb{R}^+)} F)(\mathbf{x}', h') = F(h^{-1} \odot \mathbf{x}', h^{-1} \cdot h') = F(\tfrac{1}{s}\mathbf{x}', \tfrac{s'}{s}).$$

Figure 7: **Top row**: In group theoretical terms we can describe a smiley face via a collection of group elements $\mathcal{S} \subset \mathbb{R}^2 \rtimes \mathbb{R}^+$, which e.g. represents the locations and scale of low-level features such as blobs/circles. Such a collection of points transforms via left multiplication, e.g. $h \cdot \mathcal{S} := \{(\mathbf{0}, h) \cdot g \mid g \in \mathcal{S}\}$, with $h \in \mathbb{R}^+$. **Bottom row**: In a group convolutional setting we can work with convolution kernels which are functions on $\mathbb{R}^2 \times \mathbb{R}^+$ that assign weights to locations in which locally scaled features are expected. Such kernels transform via the group representation.

The scaling of a $\mathbb{R}^2 \rtimes \mathbb{R}^+$ group convolution kernel is thus achieved by a planar scaling and a logarithmic shift along the scale axis. This is illustrated in Fig. 7. The determinant of the Jacobian of this action is $|\det h| = s^d$. Using the above group structure we can write out the explicit forms for the lifting (Eq. (6)) and group correlations (Eq. (7)) as follows:

$\mathbb{R}^2 \rtimes \mathbb{R}^+$-lifting:

$$(k \tilde{\star} f)(\mathbf{x}, s) = \int_{\mathbb{R}^2} \frac{1}{s^d} k(\frac{1}{s'}(\mathbf{x}' - \mathbf{x})) f(\mathbf{x}') \mathrm{d}\mathbf{x}',$$

$\mathbb{R}^2 \rtimes \mathbb{R}^+$-correlation:

$$(K \star F)(\mathbf{x}, s) = \int_{\mathbb{R}^2} \int_{\mathbb{R}^+} \frac{1}{s^d} K(\frac{1}{s'}(\mathbf{x}' - \mathbf{x}), \frac{s}{s'}) F(\mathbf{x}', \theta') \mathrm{d}\mathbf{x}' \mathrm{d}s'.$$

The logarithmic map on $H$ is provided by the natural logarithm as follows

$$\mathrm{Log}\, h = \ln s.$$

The B-spline basis, centered around each $h_i = (s_i) \in H$ with scale $s_i \in \mathbb{R}$, as depicted in Fig. 3, is thus computed via

$$B^{\mathbb{R},n} \left( \frac{\mathrm{Log}\, h_i^{-1} \cdot h}{s_h} \right) = B^{\mathbb{R},n} \left( \frac{\ln s_i^{-1} s}{s_h} \right).$$

### B.4 THE 3D ROTATION GROUP $H = SO(3)$

The 3D rotation group is given by space of $3 \times 3$ orthogonal matrices with determinant 1, with the group product and inverse given by matrix product and matrix inverse:

$$h \cdot h' = (\mathbf{R}.\mathbf{R}')$$
$$h^{-1} = \left( \mathbf{R}^{-1} \right).$$

The action of $SO(3)$ on $\mathbb{R}^3$ is given by matrix-vector multiplication

$$h \odot \mathbf{x} = \mathbf{R}.\mathbf{x},$$

with $\mathbf{x} \in \mathbf{R}^3$. Combining the group structure of 3D translations with rotations in $SO(3)$ as a semi-direct product group gives us the roto-translation group $SE(3) = \mathbb{R}^3 \rtimes SO(3)$, also known as the 3D special Euclidean motion group. The left-regular representation are then

$$(\mathcal{L}_h^{SO(3) \to \mathbb{L}_2(\mathbb{R}^3)} f)(\mathbf{x}') = f(h^{-1} \odot \mathbf{x}') = f(\mathbf{R}^{-1}.\mathbf{x}'),$$
$$(\mathcal{L}_h^{SO(3) \to \mathbb{L}_2(SO(3))} F)(h') = F(h^{-1} \cdot h') = F(\mathbf{R}^{-1}.\mathbf{R}'),$$
$$(\mathcal{L}_h^{SO(3) \to \mathbb{L}_2(SE(3))} F)(\mathbf{x}', h') = F(h^{-1} \odot \mathbf{x}', h^{-1} \cdot h') = F(\mathbf{R}^{-1}.\mathbf{x}', \mathbf{R}^{-1}.\mathbf{R}').$$

The determinant of the Jacobian of the action of $SO(3)$ on $\mathbb{R}^3$ is $|\det h| = 1$. Using the above group structure we can write out the explicit forms for the lifting (Eq. (6)) and group correlations (Eq. (7)) as follows:

$SE(3)$-lifting:
$$(k \tilde{\star} f)(\mathbf{x}, \mathbf{R}) = \int_{\mathbb{R}^2} k(\mathbf{R}^{-1}.(\mathbf{x}' - \mathbf{x})) f(\mathbf{x}') \mathrm{d}\mathbf{x}',$$

$SE(3)$-correlation:
$$(K \star F)(\mathbf{x}, \mathbf{R}) = \int_{\mathbb{R}^2} \int_{SO(3)} K(\mathbf{R}^{-1}.(\mathbf{x}' - \mathbf{x}), \mathbf{R}^{-1}.\mathbf{R}') F(\mathbf{x}', \mathbf{R}') \mathrm{d}\mathbf{x}' \mathrm{d}\mathbf{R}',$$

with $\mathrm{d}\mathbf{R}'$ denoting the Haar measure on $SO(3)$, which depends on the parameterization, see details below.

The logarithmic map from the group $SO(3)$ to the Lie algebra $\mathfrak{so}(3)$ is given by the matrix logarithm and the resulting matrix can be expanded in a basis $\{A_1, A_2, A_3\}$ for the Lie algebra

$$\mathrm{Log}\, \mathbf{R} = \sum_{i=1}^3 a^i A_i,$$

with

$$A_1 = \begin{pmatrix} 0 & 0 & 0 \\ 0 & 0 & -1 \\ 0 & 1 & 0 \end{pmatrix}, \quad A_2 = \begin{pmatrix} 0 & 0 & 1 \\ 0 & 0 & 0 \\ -1 & 0 & 0 \end{pmatrix}, \quad A_3 = \begin{pmatrix} 0 & -1 & 0 \\ 1 & 0 & 0 \\ 0 & 0 & 0 \end{pmatrix},$$

which at the origin represent an infinitisimal rotation around the $x$, $y$, and $z$ axis respectively. A cardinal B-spline centered at some $\mathbf{R}_i \in SO(3)$ with scale $s_h$ can then be computed in terms of these coefficients via $B^{\mathbb{R},n} \left( \frac{\mathrm{Log}\, \mathbf{R}_i^{-1}.\mathbf{R}}{s_h} \right)$.

In practice it is often convenient to rely on a parameterization of the group and define the group structure in terms of these parameters. A common choice is to do this via ZYZ Euler angles via

$$\mathbf{R}_{\alpha, \beta, \gamma} = \mathbf{R}_{\mathbf{e}_z, \gamma}.\mathbf{R}_{\mathbf{e}_y, \beta}.\mathbf{R}_{\mathbf{e}_z, \alpha},$$

with $\mathbf{R}_{\mathbf{e}, \theta}$ a rotation of $\theta$ around a reference axis $\mathbf{e}$, and $\alpha \in [0, 2\pi], \beta \in [0, \pi], \gamma \in [0, 2\pi]$. A Haar measure in terms of this parameterization is then given by $\mathrm{d}\mu(R) = \sin \beta \mathrm{d}\alpha \mathrm{d}\beta \mathrm{d}\gamma$. We will use this parameterization in construction of the quotient group $SO(3)/SO(2)$ next.

## B.5 THE 2-SPHERE $H = S^2 \equiv SO(3)/SO(2)$

Here we define the 2-sphere as a group quotient of $SO(3)$ and remark that the same equations as in the $SO(3)$ case (App. B.4) are used for the lifting and group correlations. This section describes how to construct convolution kernels on the sphere using the logarithmic map of $SO(3)$, which can then be used to build G-CNNs on $\mathbb{R}^d \rtimes S^2$.

The 2-sphere is defined as $S^2 = \{\mathbf{x} \in \mathbb{R}^3 \mid \|\mathbf{x}\| = 1\}$. Any point on the sphere can be obtained by rotating a reference vector $\mathbf{z} = (0, 0, 1)^T$ with elements of $SO(3)$, i.e., $\forall_{\mathbf{n} \in S^2}, \exists_{\mathbf{R} \in SO(3)} : \mathbf{n} = \mathbf{R}.\mathbf{z}$. In other words, the group $SO(3)$ acts transitively on $S^2$. In ZYZ Euler angle parameterization of $SO(3)$ all angles $\alpha$ leave the reference vector $\mathbf{z}$ in place, meaning that for each $\mathbf{n} \in S^2$ we have several $\mathbf{R} \in SO(3)$ that map $\mathbf{z}$ to the same $\mathbf{n}$. As such, we can treat $S^2$ as the quotient group $SO(3)/SO(2)$, where $SO(2)$ refers to the sub-group of rotations around the $z$-axis.

In order to define B-splines on the 2-sphere we need a logarithmic map from a point in $S^2$ to the (Euclidean) tangent vector space $T_e(S^2)$ at the origin. We will construct this logarithmic map using the Log define for $SO(3)$. Let us parameterize the sphere with

$$\mathbf{n}(\beta, \gamma) = \mathbf{R}_{\mathbf{e}_z, \gamma}.\mathbf{R}_{\mathbf{e}_y, \beta}.\mathbf{z}.$$

Any rotation $\mathbf{R}_{\alpha, \beta, \gamma}$ with arbitrary $\alpha$ maps to the same $\mathbf{n}(\beta, \gamma) \in S^2$. As such, there are also many vectors $A = \text{Log } \mathbf{R}_{\alpha, \beta, \gamma} \in T_e(SO(3))$ that map to a suitable rotation matrix via the exponential map $\mathbf{R} = \exp A$. We aim to find the vector in $T_e(SO(3))$ for which $c^3 = 0$, which via the exponential map generate torsion free exponential curves. The Log of any $\mathbf{R}_{\alpha, \beta, \gamma}$ with $\alpha = -\gamma$ results in such a vector (Portegies et al., 2015). As such we define

$$\text{Log}_{S^2}\, \mathbf{n}(\beta, \gamma) := \text{Log}_{SO(3)}\, \mathbf{R}_{-\gamma, \beta, \gamma},$$

which maps any point in $S^2$ to a 2-dimensional vector space $T_e(S^2) \subset T_e(SO(3))$. A B-spline on $S^2$ can then be defined via

$$f(\beta, \gamma) = \sum_{i=1}^{N} c^i B^{\mathbb{R}, n} \left( \frac{\text{Log}_{S^2}\, \mathbf{R}_{0, \beta_i, \gamma_i}^{-1}.\mathbf{R}_{0, \beta, \gamma}.\mathbf{z}}{s_h} \right), \tag{16}$$

in which individual splines basis functions are centered around points $\mathbf{n}(\beta_i, \gamma_i)$.

We remark that the group product $\mathbf{R}_{\alpha_i, \beta_i, \gamma_i}^{-1}.\mathbf{R}_{0, \beta, \gamma}$ generates different rotations when varying $\alpha_i$, that however still map to the same $\mathbf{n}$. The vectors obtained by taking $\text{Log}_{S^2}$ of the rotation matrices rotate with the choice for $\alpha_i$. Since the B-splines are approximately isotropic we neglect this effect and simply set $\alpha_i = 0$ in Eq. (16). Finally, we remark that the superposition of shifted splines (as in Eq. (16)) is not isotropic by construction, which is desirable when using the spline as a convolution kernel to lift functions to $SO(3)$. When constraining G-CNNs to generate feature maps on $S^2$, the kernels are constrained to be isotropic. Alternatively on could stay on $S^2$ entirely and resort to gauge-equivariant networks (Cohen et al., 2019), for which the proposed splines are highly suited to move from the discrete setting (as in (Cohen et al., 2019)) to the continuous setting, see also App. C.2. For examples of splines on $S^2$ see Figs. 3 and 4.

# C  RELATED WORK

## C.1  DEEP SCALE-SPACS

### C.1.1  SCALE SPACE LIFTING AND CORRELATIONS

In (Worrall & Welling, 2019) images $f \in \mathbb{L}_2(\mathbb{R}^d)$ are lifted to a space of positions and scale parameters by constructing a scale space via

$$f^{\uparrow}(\mathbf{x}, s) := f_s(\mathbf{x}) := (G_s \star f)(\mathbf{x})$$

with $G_s(\mathbf{x}) = (4\pi s)^{-d/2} e^{-\frac{\|\mathbf{x}\|^2}{4s}}$. The kernels and images are sampled on a discrete grid. Let $\Omega \subset \mathbb{Z}^d$ be the support of the kernel. Then the discrete scale space correlation is given by (Worrall & Welling, 2019, Eq. (19))

$$(K \star_S f^{\uparrow})(\mathbf{x}, s) = \sum_{\tilde{\mathbf{x}} \in \Omega} \sum_{\tilde{s} \in H_d} K(\tilde{\mathbf{x}}, \tilde{s}) f_{\tilde{s}s}(s\tilde{x} + \mathbf{x}),$$

with $H_d$ the discretized set of scales, e.g., $H_d = \left\{ 2^{i-1} \right\}_{i=1}^{N_h}$, where we remark that here we use the convention of scaling of a function by $s \geq 1$ instead of using the dilation parameter $a = \frac{1}{s}$ in (Worrall & Welling, 2019). Next we remark that the scale space correlation without any scale interaction ($H_d = \{1\}$) is defined by a 2D correlation kernel via

$$(k \star_S f^{\uparrow})(\mathbf{x}, s) = \sum_{\tilde{\mathbf{x}} \in \mathbb{Z}^d} k(\tilde{\mathbf{x}}) f_s(s\tilde{\mathbf{x}} + \mathbf{x}),$$

which can be regarded as a discrete atrous/dilated correlation on each of the scale slices of $f^{\uparrow}(\mathbf{x}, s)$ with kernels dilated by a factor $s$.

### C.1.2 RELATION TO LIFTING CORRELATIONS (EQ. (6)) WITH B-SPLINES

Let our lifting correlation kernel $k$ be given in a B-spline basis via Eq. (11) and let $c : \Omega \subset \mathbb{Z}^d \to \mathbb{R}$ be the map that assigns the weights to each B-spline center $\mathbf{x}_i \in \Omega$ with $\Omega$ the set of spline centers (i.e. $c(\mathbf{x}_i) = c_i$). Let the Gaussian kernel $G_s(\mathbf{x})$ be approximated by a scaled B-spline (up to a factor) and define $B_s^{\mathbb{R}^d,n} := \frac{1}{s^d} B^{\mathbb{R}^d,n}(\frac{1}{s}\mathbf{x})$. With such an approximation (see also (Bouma et al., 2007)) our lifting correlation via Eq. (6) coincides with the lifting of (Worrall & Welling, 2019) followed by their non-scale interacting scale-space correlation, i.e.,

$$(k \tilde{\star} f)(\mathbf{x}, h) = (c \star_S f^{\uparrow})(\mathbf{x}, s),$$

with $c : \Omega \to \mathbb{R}$ the spline coefficients. We show this by rewriting

$$\begin{aligned}
(k \tilde{\star} f)(\mathbf{x}, h) &= \int_{\mathbb{R}^d} \frac{1}{|\det h|} k(h^{-1} \odot (\tilde{\mathbf{x}} - \mathbf{x})) f(\tilde{\mathbf{x}}) \mathrm{d}\tilde{\mathbf{x}} \\
&= \int_{\mathbb{R}^d} \frac{1}{s^d} \sum_{\mathbf{x}_i \in \Omega} c(\mathbf{x}_i) B^{\mathbb{R}^d,n}(\tfrac{1}{s}(\tilde{\mathbf{x}} - \mathbf{x}) - \mathbf{x}_i) f(\tilde{\mathbf{x}}) \mathrm{d}\tilde{\mathbf{x}} \\
&= \sum_{\mathbf{x}_i \in \Omega} c(\mathbf{x}_i) \int_{\mathbb{R}^d} B_s^{\mathbb{R}^d,n}(\tilde{\mathbf{x}} - (\mathbf{x} + s\mathbf{x}_i)) f(\tilde{\mathbf{x}}) \mathrm{d}\tilde{\mathbf{x}} \\
&= \sum_{\mathbf{x}_i \in \Omega} c(\mathbf{x}_i) (B_s^{\mathbb{R}^d,n} \star f)(\mathbf{x} + s\mathbf{x}_i) \\
&\approx \sum_{\mathbf{x}_i \in \Omega} c(\mathbf{x}_i) f_s(\mathbf{x} + s\mathbf{x}_i).
\end{aligned}$$

## C.2 GAUGE EQUIVARIANT NETWORKS

### C.2.1 GAUGE EQUIVARIANT CORRELATION

The following highlights commonalities between this paper and the work by Cohen et al. (2019) with respect to use of left-invariant vector fields in equivariant neural networks. Consider some Lie group $G$ with Lie algebra $\mathfrak{g} = T_e(G)$, the exponential map $\mathrm{Exp} : \mathfrak{g} \to G$, and logarithmic map $\mathrm{Log} : G \to \mathfrak{g}$. Consider the group correlation between a kernel and function $K, F : G \to \mathbb{R}$, given in Eq. (7), which for B-spline kernels $K$ with finite support $\Omega := \mathrm{supp}(K) \subset G$ reads as

$$(K \star F)(g) = \int_{g\Omega} K(g^{-1} \cdot \tilde{g}) F(\tilde{g}) \mu_G(\tilde{g}), \tag{17}$$

with $\mu_G(h)$ the Haar measure on $G$, and where write $K$ for the convolution kernel on $G$, and $\tilde{K}$ for the corresponding kernel on $\mathfrak{g}$:

$$K(g) = \vec{K}(\mathrm{Log}(g)), \tag{18}$$

Let $\vec{\Omega} := \mathrm{supp}(\vec{K}) \subset \mathfrak{g}$ be the support of $\vec{K}$. Finally let $\vec{\Omega}$ be localized such that $\mathrm{Exp}$ is a diffeomorphism (i.e., $\mathrm{Exp}(\vec{\Omega}) = \Omega$ and $\mathrm{Log}(\Omega) = \vec{\Omega}$).

Now consider the definition of gauge equivariant correlation on manifolds as given by Eq. (3) of Cohen et al. (2019) for the case of scalar functions (in which case the trivial representation $\rho(g) = 1$ is to be used). In this case integration takes place over the Lie algebra, and gauge equivariant correlation is defined by

$$(\vec{K} \tilde{\star} F)(g) := \int_{\Omega} \vec{K}(\mathbf{x}) F(\mathrm{Exp}_g(\mathbf{x})) \mathrm{d}\mathbf{x}, \tag{19}$$

with $\mathrm{d}\mathbf{x}$ the Lebesgue measure on $\mathbb{R}^d$, and where $\mathrm{Exp}_g$ denotes the exponential map from $T_g(G) \to G$. In our Lie group setting all tangent spaces can be identified with the tangent space at the origin (via the push-forward of left-multiplication) and we are able to write $\mathrm{Exp}_g := g \cdot \mathrm{Exp}\,\mathbf{x}$. In the setting of gauge equivariant CNNs as in (Cohen et al., 2019) the exponential maps are generally dependent on $g$, using a separate reference/gauge frame (basis for the tangent space) at each $g$.

### C.2.2 RELATION TO G-CORRELATIONS (EQ. (7)) WITH B-SPLINES

For Lie groups the following identity holds between group correlations with localized B-splines on the one hand, in which integration takes place over the group $G$ and elements are mapped to the algebra via $\mathrm{Log}$, and gauge equivariant correlation on the other hand, in which integration takes place on the tangent spaces and vectors in these tangent spaces are mapped to the manifolds via $\mathrm{Exp}$. In other words, given the definition of G cross-correlation in (7), denoted with $\star$, and gauge correlation in (19) or (Cohen et al., 2019, Eq. (1)), denoted with $\vec{\star}$, the two operators relate via

$$(K \star F)(g) = (\vec{K} \vec{\star} F)(g). \tag{20}$$

We show this by deriving

$$
\begin{aligned}
(K \star F)(g) &= \int_{g\Omega} K(g^{-1} \cdot \tilde{g}) F(\tilde{g}) \mathrm{d}\tilde{g} \\
&\overset{1}{=} \int_{\Omega} K(\tilde{g}) F(g \cdot \tilde{g}) \mathrm{d}\tilde{g} \\
&\overset{2}{=} \int_{\vec{\Omega}} \vec{K}(\tilde{\mathbf{x}}) F(g \cdot \mathrm{Exp}(\tilde{\mathbf{x}})) \mathrm{d}\tilde{\mathbf{x}}, \\
&= \int_{\vec{\Omega}} \vec{K}(\tilde{\mathbf{x}}) F(\mathrm{Exp}_g(\tilde{\mathbf{x}})) \mathrm{d}\tilde{\mathbf{x}} \\
&= (\vec{K} \vec{\star} F)(g).
\end{aligned}
$$

In the above $\mathrm{d}\tilde{g}$ is a Haar measure on $G$. At $\overset{1}{=}$ the substitution $\tilde{g} \to g \cdot \tilde{g}$ is made and left-invariance of the Haar measure is used $(\mathrm{d}(g \cdot \tilde{g}) = \mathrm{d}\tilde{g})$. At $\overset{2}{=}$ we switch from integration over the region $\Omega$ in the Lie group to integration over region $\vec{\Omega} = \mathrm{Log}(\Omega)$ in the Lie algebra.

## D  G-CNN ARCHITECTURES

This section describes the G-CNN architectures used in the experiments of Sec. 4 using the layers as defined in Sec. 3.2 and illustrated in Fig. 1. Two slightly different architectures are in the two different tasks (metastasis classification and landmark detection), but both are regular sequential G-CNNs that start with a lifting layer (6), followed by a several group correlation layers (7), possibly alternated with spatial max-pooling, followed by a projection over $H$ via (8), and end with a $1 \times 1$ convolution or fully connected layers. The architectures are summarized in Table. 1 and 2. Note that the output of the PCam architecture is two probabilities (1 for each class), and the output of the CelebA is five heatmaps (1 for each landmark).

### D.1  PCAM

The architecture for metastasis classification in the PCam dataset is given in Tab. 1. The input $(64 \times 64)$ is first cropped to $88 \times 88$ and is then used as input for the first layer (the lifting layer). None of the layers use spatial padding such that the image is eventually cropped to size $1 \times 1$. Each layer is followed by batch normalization[1] and a ReLU activation function, except for the last layer (layer 7) which is followed by adding a bias vector of length 2 and a softmax.

Note that the first five layers, including max pooling over rotations, encode the image into a 64-dimensional rotation invariant feature vector. The final two layers (6 and 7) can be regarded as a classical neural network classifier.

To reduce a possible orientation bias we aim to approximate the support of the kernels with a disk, rather than a rectangle. We do this by only considering splines with basis function centers $\{\mathbf{x}_i \in \mathbb{Z}^d \mid \|\mathbf{x}_i\| \leq r$, with radius $r$. For $5 \times 5$ kernels we set $r = \sqrt{5}$ by which we discard the basis functions at the corners of the $5 \times 5$ grid. The grid on $H$ is uniformly sampled with $N_h$ samples, giving the discretized grid $H_d = \{(i-1) * \frac{2\pi}{N_h}\}_{i=1}^{N_h}$. The global kernels (both dense and atrous) have

---

[1]We apply batch normalization over the domain of the feature maps, so over $X = \mathbb{R}^d$ or over $X = \mathbb{R}^d \times H$, as in (Cohen & Welling, 2016).

their centers also equidistant and globally cover $S^1$, i.e., $h_i \in \{(i-1) * \frac{2\pi}{N_k}\}_{i=1}^{N_k}$, with the scales of the dense and atrous kernels respectively given by $s_h = \frac{2\pi}{N_k}$ and $s_h = \frac{2\pi}{N_h}$. The localized kernels have their centers on the grid with $h_i \in \{i\frac{2\pi}{N_h}\}_{i=-\lfloor N_k/2\rfloor}^{\lfloor N_k/2\rfloor}$ and have scale $s_h = \frac{2\pi}{N_h}$.

Finally, we follow the same data-augmentations at train time as proposed in (Liu et al., 2015). These include geometric augmentations such as $90°$ rotations and horizontal flips, as well as color augmentations such as brightness, saturation, hue and contrast variations.

Table 1: PCam $SE(2)$ G-CNN settings and the number of free parameters. Here $N_k$ denotes the number basis functions used on the $H = SO(2)$ part of the group correlation kernels.

| Basis size: | $N_k = 1$ | $N_k = 3$ | $N_k = 4$ | $N_k = 5$ | $N_k = 8$ | $N_k = 12$ | $N_k = 16$ |
|---|---|---|---|---|---|---|---|
| Layer | Nr of output feature maps (# weights) | | | | | | |
| 1: Lifting $(5 \times 5)$ | 40 (2,520) | 23 (1,449) | 20 (1,260) | 18 (1,134) | 14 (882) | 11 (693) | 10 (630) |
| Spatial max pooling $(2 \times 2)$ by a factor 2 | | | | | | | |
| 2: G-corr $(5 \times 5 \times N_h)$ | 40 (33,600) | 23 (33,327) | 20 (33,600) | 18 (34,020) | 14 (32,928) | 11 (30,492) | 10 (33,600) |
| Spatial max pooling $(2 \times 2)$ by a factor 2 | | | | | | | |
| 3: G-corr $(5 \times 5 \times N_h)$ | 40 (33,600) | 23 (33,327) | 20 (33,600) | 18 (34,020) | 14 (32,928) | 11 (30,492) | 10 (33,600) |
| Spatial max pooling $(3 \times 3)$ by a factor 3 | | | | | | | |
| 4: G-corr $(5 \times 5 \times N_h)$ | 40 (33,600) | 23 (33,327) | 20 (33,600) | 18 (34,020) | 14 (32,928) | 11 (30,492) | 10 (33,600) |
| 5: G-corr $(1 \times 1 \times N_h)$ | 64 (2,560) | 64 (4,416) | 64 (5,120) | 64 (5,760) | 64 (7,178) | 64 (8,448) | 64 (10,240) |
| Max pooling over $H$ (Projection layer) | | | | | | | |
| 6: 2D-corr $(1 \times 1)$ | 16 (1,024) | 16 (1,024) | 16 (1,024) | 16 (1,024) | 16 (1,024) | 16 (1,024) | 16 (1,024) |
| 7: 2D-corr $(1 \times 1)$ | 2 (32) | 2 (32) | 2 (32) | 2 (32) | 2 (32) | 2 (32) | 2 (32) |
| Softmax | | | | | | | |
| Total # weights: | 106,936 | 106,902 | 108,236 | 110,010 | 107,890 | 101,673 | 112,726 |

## D.2 CELEBA

The architecture for landmark detecion in the CelebA dataset is biven in Tab. 2. The input is formatted according to the details in Sec. 4. In each layer zero padding is used in order to map the $128 \times 128$ input images to a $128 \times 128$ output heatmaps. Each layer is followed by batch normalization and a ReLU activation function, except for the last layer (layer 10) which is followed by adding a bias vector and a logistic sigmoid activation function.

Note that the result of the first 6 layers, including average pooling over scale, assign locally scale-invariant feature vectors to each pixel. The final layers convert these feature maps into heatmaps via regular 2D convolutions.

Landmarks are localized via the argmax on each heatmap. The results in Fig. 5 show the success rate for localizing a landmark correctly. The success rate is computed as the average fraction of successful detections for all five landmarks in all images. A detection is considered successful if the distance to the actual landmark is less then 10 pixels.

The $H$ axis is uniformly sampled (w.r.t. to the metric on $H$) on a fixed scale range, generating the discrete sets $H_d = \{e^{(i-1)s_h}\}_{i=1}^{N_h}$ with $s_h = \frac{1}{2}\ln 2$. The global kernels have their centers on this grid, i.e., $h_i \in H_d$ with the scale parameter the same as that of the grid. The local kernels also have their centers equidistant (with scale $s_h$) to eachother, but are localized and given by $h_i \in \{e^{is_h}\}_{i=-\lfloor N_k/2\rfloor}^{\lfloor N_k/2\rfloor}$.

Table 2: CelebA scale-translation G-CNN settings and the number of free parameters. Here $N_k$ denotes the number basis functions used on the $H = (\mathbb{R}^+, \times)$ part of the group correlation kernels.

| Basis size: | $N_k = 1$ | $N_k = 2$ | $N_k = 3$ | $N_k = 4$ | $N_k = 5$ |
|---|---|---|---|---|---|
| Layer | Nr of output feature maps (# weights) | | | | |
| 1: Lifting ($5 \times 5$) | 27 (2,025) | 21 (1,575) | 17 (1,275) | 15 (1,125) | 14 (1,050) |
| 2: G-corr ($5 \times 5 \times N_h$) | 27 (18,225) | 21 (22,050) | 17 (21,675) | 15 (22,500) | 14 (24,500) |
| 3: G-corr ($5 \times 5 \times N_h$) | 27 (18,225) | 21 (22,050) | 17 (21,675) | 15 (22,500) | 14 (24,500) |
| Average pooling over $H$ (Projection layer) | | | | | |
| Spatial max pooling ($2 \times 2$) by a factor 2 | | | | | |
| 4: Lifting ($5 \times 5$) | 27 (18,225) | 21 (11,025) | 17 (7,225) | 15 (5,625) | 14 (4,900) |
| 5: G-corr ($5 \times 5 \times N_h$) | 27 (18,225) | 21 (22,050) | 17 (21,675) | 15 (22,500) | 14 (24,500) |
| 6: G-corr ($5 \times 5 \times N_h$) | 27 (18,225) | 21 (22,050) | 17 (21,675) | 15 (22,500) | 14 (24,500) |
| Average pooling over $H$ (Projection layer) | | | | | |
| Up-sampling by a factor 2 | | | | | |
| 7: 2D-corr ($3 \times 3$) | 32 (15,552) | 32 (12,096) | 32 (9,792) | 32 (8,640) | 32 (8,064) |
| 8: 2D-corr ($3 \times 3$) | 32 (9,216) | 32 (9,216) | 32 (9,216) | 32 (9,216) | 32 (9,216) |
| 9: 2D-corr ($1 \times 1$) | 64 (2048) | 64 (2048) | 64 (2048) | 64 (2048) | 64 (2048) |
| 10: 2D-corr ($1 \times 1$) | 5 (320) | 5 (320) | 5 (320) | 5 (320) | 5 (320) |
| Logistic sigmoid | | | | | |
| Total # weights: | 120,286 | 124,480 | 116,576 | 116,974 | 123,589 |

