# OpenReview forum: "B-Spline CNNs on Lie groups"
_ICLR.cc/2020/Conference — Accept (Poster)_

### Official Review · AnonReviewer1 · 2019-10-23
**Official Blind Review #1**

**Rating:** 8

**Review:**

In this paper, a framework for building group CNN with an arbitrary Lie group G is proposed. Generally, such a group CNN consists of 3 types of layers: a lifting layer which lifts a 2D image to a 3D data (G-image) whose domain is G; a group correlation layer which computes a 3D G-image from a 3D G-image; and a projection layer from a 3D G-image to a 2D image. To implement the convolutions in the lifting layer and group correlation layer which are defined in the continuous setting, the B-Spline basis functions are applied to expand the convolution kernels. Experimental results on tumor clarification and landmark localization show the superiority over CNN.

Advantages:
1. A flexible framework for group convolutional neural network is proposed with strong theoretical support in Theorem 1.
2. Familiar properties of convolutions from classical CNN design (like localized, atrous, and deformable convolutions) can also be implemented in G-CNN using specified B-Spline basis functions.
3. In comparison with standard CNN, the effectiveness of the B-Spline-based G-CNN is validated through experiments on two typical data sets.

Weakness:
1. [Readability] For readers who are not familiar with Lie groups, this paper is very hard to follow.
(1)	For Theorem 1, the authors are suggested to give some illustrative explanation. Besides, what is “Stab_G”?
(2)	The architecture of G-CNN, i.e., the 3 types of layers, are directly given in Eqs. (5)-(7) without examples, illustrative examinations, or visual illustrations.
(3)	Fig. 1 can be modified for better readability.

2. [Experiments] The proposed G-CNN has some similarities with data augmentation (like rotation, scaling) based CNN. Then, how better can the G-CNN perform than CNN with data augmentation? More experiments on this point are suggested, and relevant theoretical explanations will be appreciated.

3. [Implementation] Considering the complicated mathematics in this paper, I am afraid that implementation of the proposed G-CNN is also very hard. It would be better for the authors to discuss the implementation. In my mind, if the implementation is not so hard, then the formulation of G-CNN can also be simplified for better readability.


**Experience Assessment:**

I have read many papers in this area.

**Review Assessment: Checking Correctness Of Derivations And Theory:**

I assessed the sensibility of the derivations and theory.

**Review Assessment: Checking Correctness Of Experiments:**

I assessed the sensibility of the experiments.

**Review Assessment: Thoroughness In Paper Reading:**

I read the paper at least twice and used my best judgement in assessing the paper.

---

> ### Author Response · Authors · 2019-11-08
> **First thoughts**
>
> Thank you for such a careful analysis of the paper! We also thank you for identifying some points for improvement; we address these in our revision and believe it leads to a much improved paper. We discuss them below. We are currently working on updating the manuscript. If in the meantime you have additional questions we would be happy to respond to them!
>
> ***
> “[Readability] For readers who are not familiar with Lie groups, this paper is very hard to follow.
> (1) For Theorem 1, the authors are suggested to give some illustrative explanation. Besides, what is “Stab_G”?
> (2) The architecture of G-CNN, i.e., the 3 types of layers, are directly given in Eqs. (5)-(7) without examples, illustrative examinations, or visual illustrations.
> (3) Fig. 1 can be modified for better readability. “
>
> (1) We are working on an illustration of theorem 1 for the case of roto-translation equivariant networks, and will place this in appendix B.2 and refer to it in the main text. We will provide extra explanation for each layer to give a more context.
> (2) We will subsequently add a paragraph in which equations (5)-(7) are given explicitly for the roto-translation group and write out the equations for several other groups in Appendix B.
> (3) We are working on an improved introductory figure.
> All in all these modifications will probably add another page to the main body of the paper, but of course we still aim to stay within the 10 page limit. Stay tuned for the revision.
>
> ***
> “[Experiments] The proposed G-CNN has some similarities with data augmentation (like rotation, scaling) based CNN. Then, how better can the G-CNN perform than CNN with data augmentation? More experiments on this point are suggested, and relevant theoretical explanations will be appreciated. “
>
> We initially left out discussions regarding augmentation as these are addressed in prior work on G-CNNs, but we realize that it is a too important connection to ignore. So we are currently trying to find a way to fit this into the revision.
>
> There are mainly two arguments why G-CNNs are preferred over data-augmentations:
> 1. Data augmentations transform the inputs globally and are not able to deal with local transformations/symmetries. G-CNNs handle both local and global symmetries.
> 2. By using data augmentations you let the network learn how to deal with such transformations. It thus has to spend valuable network capacity on this. G-CNNs on the other hand have the appropriate geometric structure encoded in them and therefore do not have to spend valuable network capacity on learning geometric behavior, but rather can spend it all on learning effective representations.
>
> We do remark however, that data-augmentations and G-CNNs happily live together, and that data augmentations can still be used to improve performance, in particular when the augmentations include transformations that are not covered by the Lie group.
>
> ***
> "[Implementation] Considering the complicated mathematics in this paper, I am afraid that implementation of the proposed G-CNN is also very hard. It would be better for the authors to discuss the implementation. In my mind, if the implementation is not so hard, then the formulation of G-CNN can also be simplified for better readability. “
>
> In order to achieve a generic viewpoint on equivariance we make an abstraction step (and speak of representations of groups), and this step is indeed somewhat mathematically demanding, but it eventually allows us to develop the code in a modular (object oriented) and generic way. The specific equations for Lie group CNNs layers, e.g. for roto-translation equivariance, are however very readable and similar to the conventional convolution operators. We will provide such explicit examples in the revision in Appendix B, but we are trying to fit a concrete example in the main body of the text as well for the revision.
>
> Via the abstractions made in this paper a developer/researcher interested in implementing G-CNNs for a particular transformation group only has to define the group structure of the sub-group H that he/she wants to combine with translations (e.g. to build translation+rotation networks, translation+scalings networks, translations+skewing networks,…) and all the layers are automatically derived.
>
> We hope to be able to convince you of the tractability of implementing the theory by anonymously providing examples of implementations for the 2D roto-translation and scale-translation groups (as python classes), together with the g-splinets (as it is currently called) tensorflow library via the link above (here on openreview.net). We are working on this give an update when we submit the final revision. The code will appear on GitHub after the accept/reject decision is made, with minimal working examples and the script used to generate the results.

---

> > ### Author Response · Authors · 2019-11-15
> > **First thoughts (continued)**
> >
> > P.s. With respect to implementation challenges: Here is a code snipped (also promised to Rev1) that illustrates the type of coding that needs to be done. See code link above for more detail.
> >
> > ** In a “group class” file “SE2.py” we define:
> >
> > class H:
> >    # Group product: two rotation angles simply add up
> >    def prod( h_1, h_2 ):
> >               return h_1 + h_2
> >    # Group inverse: rotation angle changes sign
> >    def inv( h ):
> >  	      return –h
> >    # Logarithmic map: mapping the angle to the interval [0,2pi]
> >    def log( h ):
> >  	      return tf.mod(h + np.pi, 2*np.pi) - np.pi
> >    # The action on Rn: describes a rotation of the coordinate grid
> >    # The input is a transformation parameter h, and a coordinate grid xx. The output is the transformed coordinate grid.
> >    def left_action_on_Rn( h, xx ):
> >               x = xx[...,0]
> >               y = xx[...,1]
> >               th = h[0]
> >               x_new = x * tf.cos(th) - y * tf.sin(th)
> >               y_new = x * tf.sin(th) + y * tf.cos(th)
> >               # Reformat c
> >               xx_new = tf.stack([x_new,y_new],axis=-1)
> >               # Return the result
> >               return xx_new
> >
> > ** In the main file used to build the architecture the library is called via:
> >
> > group_name = 'SE2'
> > group = importlib.import_module('gsplinets.group.'+group_name)
> > layers = gsplinets.layers(group)
> > ...
> >
> > # Lifting layer:
> > tensor = inputs
> > l1 = layers.ConvRnG( tensor, N_out, k_size, h_grid)
> > tensor = tf.nn.relu(l1.outputs)
> > # G-conv layer
> > l2 = layers.ConvGG( tensor, N_out, k_size, h_grid)
> > tensor = tf.nn.relu(l2.outputs)
> > ...

---

### Official Review · AnonReviewer2 · 2019-10-25
**Official Blind Review #2**

**Rating:** 3

**Review:**

The paper proposes an (approximately) equivariant neural network architecture for data lying on homogeneous spaces of Lie groups. In contrast to the Gauge equivariant and Fourier approaches that have recently appeared, here the authors simply put a B-spline basis on local patches of the homogeneous space and move the basis elements around explicitly by applying the group action.

The approach is appealing in its simplicity and generality. No need to worry about irreducible representations and Fourier transforms, the formalism works for virtually any Lie group, no problem with non-compact groups. However, there is a constant need for interpolation. What is more more significant is that both the homogeneous space and the group need to be discretized and in general that cannot be done in a regular manner (no notion of a uniform grid on SO(3) for example). The authors assure us that "we find that it is possible to find approximately uniform B-splines... e.g. by using a repulsion model". I am not sure that it is so simple. This is one of those things where the idea is straightforward but the devil is in the details.

Theorem 1 seems important but it is a bit cryptic. What is the statement "a kernel satisfying such and such properties gives rise to an equivariant CNN"? Or "A CNN is equivariant if and only the kernel satisfies such and such properties"?

Concerningly, the paper is closely related to a few other papers using the spline CNN idea or at least the idea of taking a fixed set of functions and moving it around on the homogeneous space by acting on it with select group elements, most notably "Roto-translational convolutional neural networks for medical image analysis" by Bekkers et al.. The main difference of the present paper relative to that one is that the idea is fleshed out in a little more detail and is generalized from SE(2) to arbitrary Lie groups. However, conceptually there is little that is new.

In such a situation it would be important to present convincing experiments. Unfortunately in the present paper, results are only presented on 2 datasets, and the algorithm is basically only compared to different versions of itself, rather than state of the art competitors.

The paper is clearly written but the intuitive nature of the core ideas could be better conveyed e.g. by fancy diagrams.

**Experience Assessment:**

I have published one or two papers in this area.

**Review Assessment: Checking Correctness Of Derivations And Theory:**

I assessed the sensibility of the derivations and theory.

**Review Assessment: Checking Correctness Of Experiments:**

I assessed the sensibility of the experiments.

**Review Assessment: Thoroughness In Paper Reading:**

I read the paper thoroughly.

---

> ### Author Response · Authors · 2019-11-08
> **First thoughts**
>
> Thank you for your thorough analysis of the paper and for raising points for discussion which we are happy to address in the following. We are currently working on updating the manuscript. If in the meantime you have additional questions we would be happy to respond to them!
>
> ***
> “In contrast to the Gauge equivariant and Fourier approaches that have recently appeared, here the authors simply put a B-spline basis on local patches of the homogeneous space and move the basis elements around explicitly by applying the group action. “
>
> We provided a detailed discussion about the connection of this work to the theory of gauge equivariant CNNs in appendix C.2 and summarized this in the introduction. It turns out that the two viewpoints are equivalent in certain settings: we choose the gauge frames to be left-invariant vector fields generated by the Lie group structure. In a related way as is done in our paper, gauge equivariant CNNs also “simply move a kernel around” and align it with a particular vector field (gauge field). In the gauge paper, however, a particular grid/manifold is chosen that allows for discrete convolutions and as such avoid interpolation. In this respect, we prefer to invert the “in contrast to … simply…” statement, and remark that in order to apply the gauge CNN framework to other cases (such as meshes or manifolds in general), one has at some point to resort (analytic) kernel representations that can be sampled at arbitrary points on the manifold. The proposed B-splines enable that. We agree that they are simple functions, but that is precisely why they are nice to work with.
>
> Fourier methods are a different story. These are also wonderful techniques that do not necessarily require a specific discretization grid. I would say that such methods are your method of choice when dealing with compact (unimodular) groups, but these methods do not generalize well to other types of manifolds.
>
> The purpose of this paper is to explore new ways to represent data and build learning architectures. A particular result is that in the B-spline Lie G-CNN viewpoint we can adopt conventional engineering heuristics such as working with localized, deformable and atrous convolutions, which is simply not possible in a Fourier basis.
>
> ***
> “However, there is a constant need for interpolation. What is more more significant is that both the homogeneous space and the group need to be discretized and in general that cannot be done in a regular manner (no notion of a uniform grid on SO(3) for example). The authors assure us that "we find that it is possible to find approximately uniform B-splines... e.g. by using a repulsion model". I am not sure that it is so simple. This is one of those things where the idea is straightforward but the devil is in the details. “
>
> We are a big fan of Fourier methods and irreps to steer convolution kernels (w.r.t. trafo parameters), they allow to work exclusively with the coefficients without ever having to sample them. This, however, requires specialized activation functions (several are proposed e.g. in the works by Worrall et al. 2017, Weiler et al. 2018a, Kondor 2018 and others alike). Again, these methods work well on rotation groups, but do not generalize well to other groups.
>
> Interestingly, however, in popular techniques for spherical convolutions (both Cohen 2018b and Esteves et al 2018a) one does in fact rely on sampling of the data on the sphere (with grids that are non-uniform). They rely on a sequence of spherical harmonic fits, exact convolutions in “Fourier” domain, followed by sampling again on the sphere such that element-wise nonlinearities can be applied in a conventional way. They are highly effective despite the fact that after applying such nonlinearities (1) the functions leave the spherical harmonic basis in which they were expressed and (2) the networks are not fully equivariant anymore due to the non-uniform grid. As in many real world applications one has to make a trade-off between mathematical beauty and computational efficiency or pragmatism.
>
> Regarding discretizations on uniform grids. As remarked in the main body of the paper, uniform local grids can always be constructed on Lie groups. However, on compact groups one has to be careful that the grid does not start to overlap with itself, as can happen with SO(d). Luckily on compact groups repulsion models also always work as due to the periodic nature one has that the repulsing forces do not send elements outside of the domain.
>
> Finally, in response to “the constant need for interpolation”. We do not regard the need for interpolation as a limitation. Computationally, interpolation (in our case actually just sampling of the kernels) only occurs with every transformation in the sub-group H that is sampled, and only on for the convolution kernels.

---

> > ### Author Response · Authors · 2019-11-08
> > **First thoughts (continued)**
> >
> > ***
> > “Theorem 1 seems important but it is a bit cryptic. What is the statement "a kernel satisfying such and such properties gives rise to an equivariant CNN"? Or "A CNN is equivariant if and only the kernel satisfies such and such properties"? “
> >
> > These are excellent questions and raise a valid concern regarding the readability. We will improve the presentation by elaborating on the theorem in the text and by adding additional illustrations in appendix B where concrete examples are discussed (also in response of reviewer 3). The summary is as follows. In CNNs we work with feature maps and transformations between them. In general these layers can be very complicated and are described by two-argument kernel operators. But if we want to constrain such layers to be equivariant w.r.t. translations then it turns out that we are only allowed to use group convolutions which are fully described by only a single-argument convolution kernel.
> >
> > In image analysis we are used to working with 2D feature maps (functions on X=R^2). Now the theorem says that if we want to stick to working with 2D feature maps (X=Y=R^2) and want to have equivariance w.r.t. not just translations, but also rotations (so to SE(2)), then our only option is to work with isotropic (Eq. 4) convolution kernels (since $\mathbb{R}^2 \equiv SE(2)/SO(2)$). If we do not want to have any isotropy constraints on the convolution kernels, than we need to lift the data to higher dimensional feature maps (Y=SE(2)). This then defines lifting correlations.
> >
> > In general the theorem gives you a recipe for obtaining the type of layer that you are allowed to use given a choice of group to which you want to be equivariant to, and given a preferred domain on which to represent the feature maps.
> >
> > ***
> > “Concerningly, the paper is closely related to a few other papers using the spline CNN idea or at least the idea of taking a fixed set of functions and moving it around on the homogeneous space by acting on it with select group elements, most notably "Roto-translational convolutional neural networks for medical image analysis" by Bekkers et al.. The main difference of the present paper relative to that one is that the idea is fleshed out in a little more detail and is generalized from SE(2) to arbitrary Lie groups. However, conceptually there is little that is new. “
> >
> > We agree on related work, and remark that in fact the paper by Bekkers et al. inspired us to propose a comprehensive generalization of their method (also stated in the main text). We however do not agree that in the current paper the idea is just “fleshed out in a little more detail” and that “conceptually there is little that is new”. We believe that precisely on a conceptual level we made a significant contribution by realizing that splines can be defined on arbitrary Lie groups by defining them on the Lie algebra. This viewpoint is entirely unique and is in no way considered in the paper by Bekkers et al., where they were only able to construct B-splines on SE(2) using the group parameterization since the sub-group of rotations is 1-dimensional. By the proposed generalization we are able to apply the theory to a very large class of problems that do not just involve rotations. The fact that we can now do this is both theoretically as well as practically demonstrated.
> >
> > ***
> > "In such a situation it would be important to present convincing experiments. Unfortunately in the present paper, results are only presented on 2 datasets, and the algorithm is basically only compared to different versions of itself, rather than state of the art competitors. "
> >
> > The paper proposes CNN layers that can be used in any CNN architecture. As such, the purpose of the experiments is not to outperform any of those architectures in literature (which to choose?) but rather show (1) that group convolutional layers should be used when equivariance is desired and (2) that we can now actually build G-CNNs (for the first time) that are not based on roto-translations (e.g. scale-translation CNNs). We believe that only by comparing the method to different versions of itself (which includes standard 2D CNN architecture design) we are able to draw sensible conclusions and gain insight in how it behaves in different settings.
> >
> > ***
> > "The paper is clearly written but the intuitive nature of the core ideas could be better conveyed e.g. by fancy diagrams."
> >
> > We agree that an intuitive exposition of the method is important (also mentioned by reviewers 1 and 3). We are working on a new introduction figure.

---

### Official Review · AnonReviewer3 · 2019-11-01
**Official Blind Review #3**

**Rating:** 6

**Review:**

This paper proposes a neural network architecture which that enables the implementation of group convolutional neural networks for arbitrary Lie groups. This lifts a significant limitation of such models which were previously confined to discrete or continuous compact groups due to tractability issues.
I'm afraid that this paper is over my head. It relies heavily on field-specific terminology and as such is likely to be accessible to a relatively small subset of researchers. This looks to me like a solid contribution, however I'm really not qualified to judge.

**Experience Assessment:**

I do not know much about this area.

**Review Assessment: Checking Correctness Of Derivations And Theory:**

I did not assess the derivations or theory.

**Review Assessment: Checking Correctness Of Experiments:**

N/A

**Review Assessment: Thoroughness In Paper Reading:**

N/A

---

> ### Author Response · Authors · 2019-11-08
> **First thoughts**
>
> Thank you for reading and for providing your high-level summary (which is correct ;)). We agree that the paper relies on advanced mathematical/geometrical concepts. We found it important to build up the proposed framework in a mathematically coherent and solid way, and the abstractions help us to make generalizations, grasp the broader picture (see also paragraphs and appendices on related work) and eventually implement the theory in an accessible, object-oriented way.
>
> Nevertheless, we also find it important that the paper is accessible to a wide audience. As such, we will open-source the code (see also the code snipped as a response to reviewer 3) and work on new figures and add extra clarifications of the theory in the main text.
>
> We are currently working on updating the manuscript. If in the meantime you have additional questions we would be happy to respond to them!

---

### Author Response · Authors · 2019-11-15
**Revised paper**

We thank the reviewers again for their time and valuable and constructive feedback. In our revision we have carefully addressed the suggestions and discussion points of each reviewer. We believe that this considerably improved the paper. Although all reviewers agree that the paper presents solid work, they also agree that paper is heavy on the math which makes it hard to read: “the intuitive nature of the core ideas could be better conveyed e.g. by fancy diagrams.” We fully agree and this has been the core focus of our revision. In addition to new figures and clarifications we also added extra experiments (G-CNNs’ relation to data-augmentation) by which we addressed questions/remarks by Rev2 and Rev3. The main changes are as follows.

** In order to improve readability of the paper and make it accessible to a wider audience we made the following modifications:
   * We put great effort in crafting a new introductory figure (Fig. 1) and believe that it intuitively illustrates the main components of G-CNNs and their relations to the part-whole/capsule viewpoint.
   * We also included a new figure (Fig. 2) that illustrates the idea of defining convolution kernels on the Lie algebra via the Log-map.
   * Additionally we added a concrete example of the group structures and the actual group convolution operators in the main body of the text, and wrote out explicit examples for several groups in the appendix B. Moreover, we added two new illustrations (Fig. 6 and 7) for the group representations, which are core components in the theory and experiments.
   * The main theorem is now better introduced and explained (if you want your networks to be equivariant, than you should use G-CNNs).
   * In several places we slightly rewrote technicalities or inserted an additional brief explanation.

** Rev3 had a related concern on whether or not the theory is too complicated to be actually implemented. We hope that the added examples and illustrations alleviate this concern. We furthermore now anonymously provide the code used in the experiments (see link above) and share an open access repository after publication.

** Rev2 had several points for discussion regarding related work and the limitations of the method. We have addressed these in detail in our first response, but we also believe that in our thorough literature study and discussions in the paper itself we already addressed these in our first submission (see e.g. app C.2 "Gauge equivariant networks").

** Rev2 expressed concerns about the method being only approximately equivariant due to discretizations. The experiments show that networks greatly benefit from (both scale and rotation) equivariance which is provided by the G-CNNs. We further experimentally addressed the equivariance property with new experiments in which we compare model training with and without rotation augmentation. From this we drew the following conclusions:
   * “… comparing the models with and without $90^\circ$ augmentation show that such augmentations are crucial for the 2D model but hardly affect the $SE(2)$ model. Moreover, the $SE(2)$ model *without* outperforms the 2D model *with* augmentation. This confirms the theory: G-CNNs guarantee both local and global equivariance by construction, whereas with augmentations valuable network capacity is spend on learning (only) global invariance. The very modest drop in the $SE(2)$ case may be due to discretization of the network on a grid after which it is no longer purely equivariant but rather approximately, which may be compensated for via augmentations.”

** Rev3 had a question on how our method relates to data-augmentation. This is answered by the above.

---

### Decision · Program_Chairs · 2019-12-19

**Decision:**

Accept (Poster)

**Comment:**

The paper describes principles for endowing a neural architecture with invariance with respect to a Lie group. The contribution is that these principles can accommodate discrete and continuous groups, through approximation via a base family (B-splines).

The main criticisms were related to the intelligibility of the paper and the practicality of the approach, implementation-wise. Significant improvements have been done and the paper has been partially rewritten during the rebuttal period.

Other criticisms were related to the efficiency of the approach, regarding how the property of invariance holds under the approximations done. These comments were addressed in the rebuttal and the empirical comparison with data augmentation also supports the merits of the approach.

This leads me to recommend acceptance. I urge the authors to extend the description and discussion about the experimental validation.